# Optimal Spectral Transitions in High-Dimensional Multi-Index Models

**Leonardo Defilippis**[1], **Yatin Dandi**[2,3], **Pierre Mergny**[2], **Florent Krzakala**[2], **and Bruno Loureiro**[1]

[1]Département d'Informatique, École Normale Supérieure, PSL & CNRS, Paris, France
[2]Information, Learning and Physics Laboratory. EPFL, CH-1015 Lausanne, Switzerland.
[3]Sloan School of Management, MIT, United States.

## Abstract

We consider the problem of how many samples from a Gaussian multi-index model are required to weakly reconstruct the relevant index subspace. Despite its increasing popularity as a testbed for investigating the computational complexity of neural networks, results beyond the single-index setting remain elusive. In this work, we introduce spectral algorithms based on the linearization of a message passing scheme tailored to this problem. Our main contribution is to show that the proposed methods achieve the optimal reconstruction threshold. Leveraging a high-dimensional characterization of the algorithms, we show that above the critical threshold the leading eigenvector correlates with the relevant index subspace, a phenomenon reminiscent of the Baik–Ben Arous–Peche (BBP) transition in spiked models arising in random matrix theory. Supported by numerical experiments and a rigorous theoretical framework, our work bridges critical gaps in the computational limits of weak learnability in multi-index model.

A popular model to study learning problems in statistics, computer science and machine learning is the *multi-index model*, where the target function depends on a low-dimensional subspace of the covariates. In this problem, one aims at identifying the $p$-dimensional linear subspace spanned by a family of orthonormal vectors $\boldsymbol{w}_{\star,1}, \ldots, \boldsymbol{w}_{\star,p} \in \mathbb{R}^d$ from $n$ independent observations $(\boldsymbol{x}_i, y_i)$ from the model:

$$y_i = g\left(\langle \boldsymbol{w}_{\star,1}, \boldsymbol{x}_i \rangle, \ldots, \langle \boldsymbol{w}_{\star,p}, \boldsymbol{x}_i \rangle\right), \qquad i \in [\![n]\!]. \tag{1}$$

This formulation encompasses several fundamental problems in machine learning, signal processing, and theoretical computer science, including: (i) Linear estimation, where $p = 1$ and $g(z) = z$, (ii) Phase retrieval, where $p = 1$ and $g(z) = |z|$. (iii) Learning Two-layer neural networks, where $p$ is the width and $g(\boldsymbol{z}) = \sum_{j \in [\![p]\!]} a_j \sigma(z_j)$ for some non-linear activation function $\sigma : \mathbb{R} \to \mathbb{R}$, or (iv) learning Sparse parity functions, where $g(\boldsymbol{z}) = \text{sign}(\prod_{j \in [\![p]\!]} z_j)$.

A classical problem in statistics [1, 2, 3], the multi-index model has recently gained in popularity in the machine learning community as a generative model for supervised learning data where the labels only depend on an underlying low-dimensional latent subspace of the covariates [4, 5, 6].

Of particular interest to this work are spectral methods, which play a fundamental role in machine learning by offering an efficient and computationally tractable approach to extracting meaningful structure from high-dimensional noisy data. A paradigmatic example is the Baik–Ben Arous–Péché (BBP) transition [7], where the leading eigenvalue of a matrix correlates with the hidden signal — a phenomenon that is ubiquitous in machine learning theory. Beyond their practical utility, spectral methods often serve as a starting point for more advanced approaches, including iterative and nonlinear techniques. This leads us to the central question of this paper:

39th Conference on Neural Information Processing Systems (NeurIPS 2025).

*Can one design optimal spectral methods that minimizes the amount of data required for identifying the hidden subspace in multi-index models?*

While the optimal spectral method for single-index models are well understood [8, 9, 10, 11], and their optimality in terms of weak recovery threshold established [12, 9] their counterparts for multi-index models remain largely unexplored. This gap is particularly important, as multi-index models serve as a natural testbed for studying the computational complexity of feature learning in modern machine learning, and have attracted much attention recently [13, 5, 6, 14, 15, 16, 17].

**Main contributions —** In this work, we step up to this challenge by constructing optimal spectral methods for multi-index models. We introduce and analyze spectral algorithms based on a linearized message-passing framework, specifically tailored to this setting. Our main contribution is to present two such constructions, establish the reconstruction threshold for these methods and to show that they achieve the provably optimal threshold for weak recovery among efficient algorithms [18].

**Other Related works –** Recently, multi-index models have become a proxy model for studying non-convex optimization [19, 14, 15]. [20] has shown that the sample complexity of one-pass SGD for single-index model is governed by the *information exponent* of the link function. This analysis was generalized in several directions, such as to other architectures [16], larger batch sizes [21] and to overparametrised models [22]. A similar notion, known as the *leap exponent*, was introduced for multi-index models, where it was shown that different index subspaces might be learned hierarchically according to their interaction [23, 13, 24, 25]. The picture was found to be different for batch algorithms exploiting correlations in the data [26, 27, 28], achieving a sample complexity closer to optimal first order methods [12, 29, 18]. [18] in particular, provided optimal asymptotic thresholds for weak recovery within the class of first order methods.

Spectral methods are widely employed as a warm start strategy for initializing other algorithms, in particular for iterative schemes (such as gradient descent) for which random initialization is a fixed point. Relevant to this work is the class of approximate message passing (AMP) algorithms, which have garnered significant interest in the high-dimensional statistics and machine learning communities over the past decade [30, 31, 32, 33]. AMP for multi-index models was discussed in [4, 18, 34, 35]. Spectral initialization for AMP in the context of single-index models has been studied by [36].

The interplay between AMP and spectral methods has been extensively studied in the literature, see for example [37, 38, 39, 9, 11, 10, 40]. In particular the idea of using message passing algorithm to derive spectral method has been very successful, leading to the non-backtracking matrix [41] and more recently to the Kikuchi hierarchy [42, 43], and to non-linear optimal spectral methods for matrix and tensor factorization [38, 44, 45, 46].

During the finalization of this work, an independent paper analysing a family of spectral estimators indexed by a pre-processing function $\mathcal{T}$ for Gaussian multi-index models [47] appeared. Their results leverage tools from random matrix theory to characterize the asymptotic spectral distribution of this family, as well as the correlation of the top eigenvectors with the indices. They prove that a tailored $\mathcal{T}$ asymptotically achieves the optimal weak recovery threshold of [18] in the particular case where $\mathbf{G}(y)$ is jointly diagonalizable for all $y$, providing an alternative proof of Conjecture 2.10 and Theorem 2.11 for this particular case.

# 1  Setting and Notations

**Notations —** To enhance readability, we adopt the following consistent notations through the paper: scalar quantities are denoted using non-bold font (e.g., $a$ or $A$), vectors are represented in bold lowercase letters (e.g., $\boldsymbol{a}$), and matrices are written in bold uppercase letters (e.g., $\boldsymbol{A}$). We further differentiate random variables depending on the noise with non-italic font (e.g, a, $\mathbf{a}$, $\mathbf{A}$). We denote by $\langle \boldsymbol{a}, \boldsymbol{b} \rangle$ the standard Euclidean scalar product, $\|\boldsymbol{a}\|$ the $\ell^2$-norm of a vector $\boldsymbol{a}$, $\|\boldsymbol{A}\|_{\mathrm{op}}$ the operator norm of a matrix $\boldsymbol{A}$ and $\|\boldsymbol{A}\|_{\mathrm{F}}$ its Frobenius norm. Given $n \in \mathbb{N}$, we use the shorthand $[\![n]\!] = \{1, \ldots, n\}$. We denote $\mathbb{S}^p_+$ the cone of positive semi-definite $p \times p$ matrices. $(x)_+ := \max(0, x)$.

**The Gaussian Multi Index Model —** We consider the supervised learning setting with $n$ i.i.d. samples $(\mathbf{x}_i, \mathbf{y}_i)_{i \in [\![n]\!]}$ with covariates $\mathbf{x}_i \sim \mathsf{N}(\mathbf{0}, d^{-1}\boldsymbol{I}_d) \in \mathbb{R}^d$ and labels y drawn conditionally from

the Gaussian *multi-index model* defined by

$$y \sim \mathsf{P}\big(.|\boldsymbol{W}_\star^T \mathbf{x}\big) = \mathsf{P}\left(\left.\cdot\right| \begin{bmatrix} \langle \boldsymbol{w}_{\star,1}, \mathbf{x} \rangle \\ \vdots \\ \langle \boldsymbol{w}_{\star,p}, \mathbf{x} \rangle \end{bmatrix}\right), \tag{2}$$

where $\boldsymbol{W}_\star := (\boldsymbol{w}_{\star,1}, \dots, \boldsymbol{w}_{\star,p}) \in \mathbb{R}^{d \times p}$ is a weight matrix with independent columns with $\boldsymbol{w}_{\star,j} \sim \mathsf{N}(\mathbf{0}, \boldsymbol{I}_d)$, $j \in \llbracket p \rrbracket$ and $\mathsf{P}(\cdot|\cdot)$ is a conditional probability distribution. Additionally, we define the *link function* $g : \mathbb{R}^p \ni \boldsymbol{z} \mapsto g(\boldsymbol{z}) \in \mathbb{R}$ as the conditional mean

$$g(\boldsymbol{z}) := \mathbb{E}\{y|\boldsymbol{W}_\star^T \mathbf{x} = \boldsymbol{z}\} = \int y \, d\mathsf{P}(y|\boldsymbol{z}), \tag{3}$$

and the labels' marginal distribution $\mathsf{Z}(y) = \mathbb{E}_{\boldsymbol{W}_\star} \mathbb{E}_{\mathbf{x} \sim \mathsf{N}(\mathbf{0}, d^{-1}\boldsymbol{I}_d)}[\mathsf{P}(y = y|\boldsymbol{W}_\star^T \mathbf{x})]$. Note that, in the limit $d \to \infty$, for $\mathbf{x} \sim \mathsf{N}(\mathbf{0}, d^{-1}\boldsymbol{I})$, the Central Limit Theorem implies $\boldsymbol{W}_\star^T \mathbf{x} \sim \mathsf{N}(\mathbf{0}, \boldsymbol{I}_p)$ and $\mathsf{Z}(y) = \mathbb{E}_{\mathbf{z} \sim \mathsf{N}(\mathbf{0}, \boldsymbol{I}_p)}[\mathsf{P}(y = y|\mathbf{z})]$.

We investigate the problem of reconstructing $\boldsymbol{W}_\star$ in the proportional *high-dimensional* limit

$$d, n \equiv n(d) \to \infty \quad \text{such that} \quad n/d \to \alpha \in \mathbb{R}_+. \tag{4}$$

In particular we are interested in the existence of an estimator $\hat{\mathbf{W}}$ that correlates with the weight matrix $\boldsymbol{W}_\star$ better than a random estimator. This is formalized as follows.

**Definition 1.1.** (Weak subspace recovery) Given an estimator $\hat{\mathbf{W}}$ of $\boldsymbol{W}_\star$ with $\|\hat{\mathbf{W}}\|_F^2 = \Theta(d)$, we say we have *weak recovery* of a subspace $V \subset \mathbb{R}^p$, if

$$\inf_{\boldsymbol{v} \in V \cap \mathbb{S}^{p-1}} \left\| \hat{\mathbf{W}}^T \boldsymbol{W}_\star \boldsymbol{v} \right\| = \Theta(1), \tag{5}$$

with high probability.

Computational bottlenecks for weak recovery in the Gaussian multi-index models have been studied by [18] using an optimal *generalized approximate message passing* (GAMP) scheme, see Appendix A for a detailed discussion of the algorithm. In particular, for the appropriate choice of denoiser functions, given in eq. (43), AMP is provably optimal among first-order methods [48, 49]. Their work provides a classification of the directions in $\mathbb{R}^p$ in terms of computational complexity of their weak learnability. In particular, if and only if

$$\mathbb{E}_{\mathbf{z} \sim \mathsf{N}(\mathbf{0}, \boldsymbol{I}_p)}[\mathbf{z}\mathsf{P}(y|\mathbf{z})] \propto \mathbb{E}[\mathbf{z}|y] = \mathbf{0} \tag{6}$$

almost surely over $y \sim \mathsf{Z}$, the subspace of directions that can be learned in a finite number of iterations is empty, for AMP randomly initialized. Nonetheless, if the initialization contains an arbitrarily small (but finite) amount of *side-information* about the ground truth weights $\boldsymbol{W}_\star$, AMP can weakly recover a subspace of $\mathbb{R}^p$, provided that $\alpha > \alpha_c$, where the critical sample complexity is characterized in Lemma 1.2.

**Lemma 1.2.** *[18], Stability of the uninformed fixed point and critical sample complexity]* *If $\boldsymbol{M} = \mathbf{0} \in \mathbb{R}^{p \times p}$ is a fixed point of the state evolution associated to the optimal GAMP (43), then it is an unstable fixed point if and only if $\|\mathcal{F}(\boldsymbol{M})\|_F > 0$ and $n > \alpha_c d$, where the critical sample complexity $\alpha_c$ is:*

$$\alpha_c^{-1} = \sup_{\boldsymbol{M} \in \mathbb{S}_+^p, \|\boldsymbol{M}\|_F = 1} \|\mathcal{F}(\boldsymbol{M})\|_F, \tag{7}$$

*with*

$$\mathcal{F}(\boldsymbol{M}) := \mathbb{E}_{y \sim \mathsf{Z}}\left[\boldsymbol{G}(y)\boldsymbol{M}\boldsymbol{G}(y)\right], \tag{8}$$

*and $\boldsymbol{G}(y) := \mathbb{E}_{\mathbf{z} \sim \mathsf{N}(\mathbf{0}, \boldsymbol{I}_p)}[\mathbf{z}\mathbf{z}^T - \boldsymbol{I}_p|y]$. Moreover, if $\|\mathcal{F}(\boldsymbol{M})\|_F = 0$, then $\boldsymbol{M} = \mathbf{0}$ is a stable fixed point for any $n = \Theta(d)$.*

The aim of our work is to close this gap, providing an estimation procedure that achieves weak recovery at the same critical threshold $\alpha_c$ defined in eq. (7), but crucially *does not require an informed initialization*. In what follows we restrict the problem defined in (3) to the set of link functions satisfying eq. (6). These functions (said to have a *generative exponent* 2 in the terminology of [29]) covers a large class of the relevant problems, with the exception of the really hard functions such as sparse parities, which cannot be solved efficiently with a linear in $d$ number of samples [18].

Throughout this manuscript we adopt the notation $\boldsymbol{a} = \text{vec}(\boldsymbol{A} \in \mathbb{R}^{b \times p})$, $b = n, d$, for the vector in $\mathbb{R}^{bp}$ with components $a_{(i\mu)} = A_{i\mu}$, $i \in [\![b]\!]$, $\mu \in [\![p]\!]$, where the double index $(i\mu) \in [\![b]\!] \times [\![p]\!]$ is a shorthand for the scalar index $i + (\mu - 1)b$. Similarly, we say that $\text{mat}(\boldsymbol{a}) = \boldsymbol{A} \iff \text{vec}(\boldsymbol{A}) = \boldsymbol{a}$.

We can now introduce the Spectral Methods we aim to investigate. Given a matrix $\mathbf{X} \in \mathbb{R}^{n \times d}$ with rows $\mathbf{x}_i \sim \mathsf{N}(\mathbf{0}, d^{-1}\boldsymbol{I}_d)$ and a vector of labels $\mathbf{y} \in \mathbb{R}^n$ with elements sampled as in (3), define $\hat{\boldsymbol{G}} \in \mathbb{R}^{np \times np}$ as

$$\hat{G}_{(i\mu),(j\nu)} := \delta_{ij} G_{\mu\nu}(\mathbf{y}_i) = \delta_{ij}\mathbb{E}_{\mathbf{z} \sim \mathsf{N}(\mathbf{0}, \boldsymbol{I}_p)}[\mathbf{z}_\mu \mathbf{z}_\nu - \delta_{\mu\nu}|\mathbf{y}_i], \tag{9}$$

and the following two spectral estimators $\hat{\mathbf{W}}_L, \hat{\mathbf{W}}_T$ of the weight matrix $\boldsymbol{W}_\star$ as

1. Asymmetric spectral method:

$$\hat{\mathbf{W}}_L := \sqrt{dN}\frac{\mathbf{X}^T\text{mat}\left(\hat{\boldsymbol{G}}\boldsymbol{\omega}_1\right)}{\|\mathbf{X}^T\text{mat}\left(\hat{\boldsymbol{G}}\boldsymbol{\omega}_1\right)\|_\mathrm{F}}, \tag{10}$$

where $\boldsymbol{\omega}_1 \in \mathbb{R}^{np}$ is the eigenvector corresponding to the eigenvalue $\gamma_1^L$ with largest real part of the matrix $\boldsymbol{L} \in \mathbb{R}^{np \times np}$ defined as:

$$L_{(i\mu),(j\nu)} := \left(\left[\mathbf{X}\mathbf{X}^T\right]_{ij} - \delta_{ij}\right) G_{\mu\nu}(\mathbf{y}_j) \quad i, j \in [\![n]\!], \mu, \nu \in [\![p]\!]. \tag{11}$$

2. Symmetric spectral method:

$$\hat{\mathbf{W}}_T := \sqrt{dN}\frac{\text{mat}(\boldsymbol{w}_1)}{\|\boldsymbol{w}_1\|}, \tag{12}$$

where $\boldsymbol{w}_1 \in \mathbb{R}^{dp}$ is the eigenvector associated to the largest eigenvalue $\gamma_1^T$ of the symmetric matrix $\boldsymbol{T} \in \mathbb{R}^{dp \times dp}$ defined as

$$T_{(k\mu),(h\nu)} := \sum_{i \in [\![n]\!]} \mathbf{X}_{ik}\mathbf{X}_{ih}\left[\boldsymbol{G}(\mathbf{y}_i)\left(\boldsymbol{G}(\mathbf{y}_i) + \boldsymbol{I}_p\right)^{-1}\right]_{\mu\nu} \quad k, h \in [\![d]\!], \mu, \nu \in [\![p]\!]. \tag{13}$$

Note that the constant $N$ is arbitrary and can be chosen to fix $d^{-1}\|\hat{\mathbf{W}}_L\|_\mathrm{F}^2 = d^{-1}\|\hat{\mathbf{W}}_T\|_\mathrm{F}^2 = N$.

At first glance, these two spectral estimators may appear ad hoc or lacking a clear theoretical justification. However, they can be motivated by a linearization of the optimal GAMP algorithm [18] around the non-informative fixed point, which reads:

$$\boldsymbol{\Omega}^t = \mathbf{X}\hat{\mathbf{W}}^t - \text{mat}\left(\hat{\boldsymbol{G}}\text{vec}\left(\boldsymbol{\Omega}^{t-1}\right)\right), \tag{14}$$

$$\hat{\mathbf{W}}^{t+1} = \mathbf{X}^T\text{mat}\left(\hat{\boldsymbol{G}}\text{vec}\left(\boldsymbol{\Omega}^t\right)\right) \tag{15}$$

Substituting the second equation into the first, this is equivalent to $\text{vec}\left(\boldsymbol{\Omega}^{t+1}\right) = \boldsymbol{L}\text{vec}\left(\boldsymbol{\Omega}^t\right)$. Moreover, assuming convergence,[1] one ends up with $\text{vec}\left(\hat{\mathbf{W}}\right) = \boldsymbol{T}\text{vec}\left(\hat{\mathbf{W}}\right)$.

This suggests that at first leading order in the estimates, the dynamics of the algorithm is governed by power-iteration (and respectively a variant of it) on the matrix $\boldsymbol{L}$ (respectively the symmetric matrix $\boldsymbol{T}$) defined previously. As power iteration converges under mild assumptions to the top (matrix) eigenvector of the tensor, this further suggests to use the top eigenvectors $\hat{\mathbf{W}}_L, \hat{\mathbf{W}}_T$ of the corresponding matrices as an estimate for the weight matrix $\boldsymbol{W}_\star$.

Additional details on the linearized GAMP are reported in Appendix A.1. Similar approaches have been thoroughly investigated in the context of single-index models [9, 10], community detection [41, 37], spiked matrix estimation [38, 39], where they have been provably shown to provide a non-vanishing correlation with the ground truth exactly at the optimal weak recovery threshold. It is interesting to notice that the spectral estimators $\hat{\mathbf{W}}_L$ and $\hat{\mathbf{W}}_T$ correspond to the generalization for multi-index models of the spectral methods derived in [10], respectively from the linearization of the optimal Vector Approximate Message Passing [50, 51] and the Hessian of the TAP free energy associated to the posterior distribution for the weights [37]. In particular, for $p = 1$, the matrices proposed in our manuscript exactly reduce to the two ones (called respectively "TAP" and "LAMP") investigated in [10].

---

[1]Here and in the rest of the paper, we drop the time index $t$ to refer to the quantities at convergence.

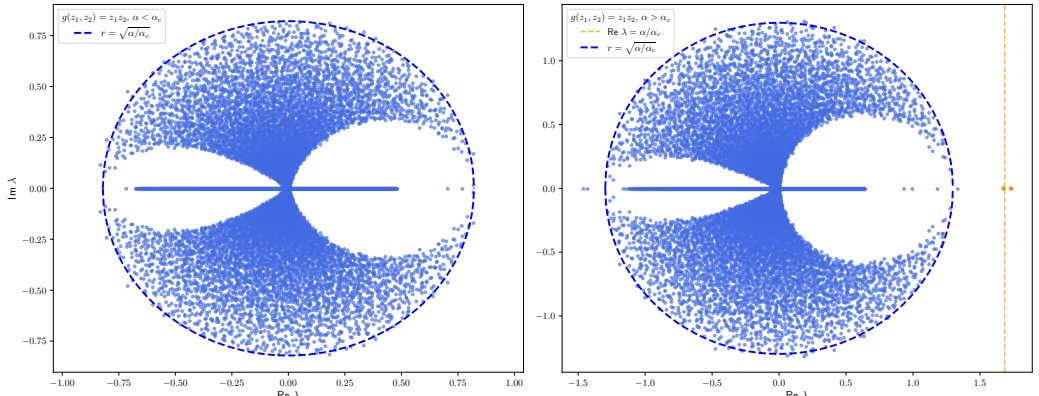

**Figure 1:** Distribution of the eigenvalues (dots) $\lambda \in \mathbb{C}$ of $\boldsymbol{L}$ at finite $n = 10^4$, for $g(z_1, z_2) = z_1 z_2$, $\alpha_c \approx 0.59375$. (**Left**) $\alpha = 0.4 < \alpha_c$. (**Right**) $\alpha = 1 > \alpha_c$. The dashed blue circle has radius equal to $\sqrt{\alpha/\alpha_c}$, *i.e.* the value $\gamma_b$ predicted in Theorem 2.4. The dashed orange vertical line corresponds to $\mathrm{Re}\,\lambda = \alpha/\alpha_c$, the eigenvalue $\gamma_s$ defined in Theorem 2.3. As predicted by the state evolution equations for this problem, two significant eigenvalues (highlighted in orange) are observed near this vertical line.

## 2 Main Technical Results

In order to characterize the weak recovery of the proposed spectral methods, we define two message passing schemes tailored to respectively have eigenvectors of $\boldsymbol{L}$ and $\boldsymbol{T}$ as fixed points. Similarly to these previous works, we leverage the *state evolution* associated to the algorithms in order to quantify the alignment between the spectral estimators and the weight matrix $\boldsymbol{W}_\star$, tracking the overlap matrices

$$\boldsymbol{M}^t \coloneqq \frac{1}{d}\left(\hat{\mathbf{W}}^t\right)^T \boldsymbol{W}_\star, \quad \boldsymbol{Q}^t \coloneqq \frac{1}{d}\left(\hat{\mathbf{W}}^t\right)^T \hat{\mathbf{W}}^t, \tag{16}$$

and their value $\boldsymbol{M}$, $\boldsymbol{Q}$, at convergence. The state evolution equations for generic linear GAMP algorithms are presented in Appendix A.2.

### 2.1 Asymmetric spectral method

**Definition 2.1.** For $\gamma > 0$, consider the linear GAMP algorithm (40,41)

$$\boldsymbol{\Omega}^t = \mathbf{X}\hat{\mathbf{W}}^t - \gamma^{-1}\mathrm{mat}\left(\hat{\boldsymbol{G}}\,\mathrm{vec}\left(\boldsymbol{\Omega}^{t-1}\right)\right), \tag{17}$$

$$\hat{\mathbf{W}}^{t+1} = \gamma^{-1}\mathbf{X}^T\mathrm{mat}\left(\hat{\boldsymbol{G}}\,\mathrm{vec}\left(\boldsymbol{\Omega}^t\right)\right). \tag{18}$$

When $\gamma$ is chosen as an eigenvalue of $\boldsymbol{L}$, the correspondent eigenvector $\boldsymbol{\omega} \in \mathbb{R}^{np}$ is a fixed point of eq. (17) for $\boldsymbol{\Omega}^t = \mathrm{mat}\,(\boldsymbol{\omega})$. In the high-dimensional limit, the asymptotic overlaps of the estimator $\hat{\mathbf{W}}$ can be tracked thanks to the state evolution equations, which follow from an immediate application of the general result of [52] to the GAMP algorithm (2.6).

**Proposition 2.2** (State evolution [52])**.** *Let $\boldsymbol{M}^t$ and $\boldsymbol{Q}^t$ denote the overlaps defined in eq. (16) for the iterative algorithm (2.1). Then, in the proportional high-dimensional limit $n, d \to \infty$ at fixed $\alpha = n/d$, they satisfy the following state evolution:*

$$\boldsymbol{M}^{t+1} = \alpha\gamma^{-1}\mathcal{F}(\boldsymbol{M}^t); \tag{19}$$

$$\boldsymbol{Q}^{t+1} = \boldsymbol{M}^t(\boldsymbol{M}^t)^T + \alpha\gamma^{-2}\left(\mathcal{G}(\boldsymbol{M}^t) + \mathcal{F}(\boldsymbol{Q}^t)\right) \tag{20}$$

*where*

$$\mathcal{F}(\boldsymbol{M}) \coloneqq \mathbb{E}_{\mathrm{y}}[\boldsymbol{G}(\mathrm{y})\boldsymbol{M}\boldsymbol{G}(\mathrm{y})], \qquad \mathcal{G}(\boldsymbol{M}) \coloneqq \mathbb{E}_{\mathrm{y}}[\boldsymbol{G}(\mathrm{y})\boldsymbol{M}\boldsymbol{G}(\mathrm{y})\boldsymbol{M}^T\boldsymbol{G}(\mathrm{y})]. \tag{21}$$

With the state evolution equations in hand, one can derive a sharp characterization of the asymptotic weak recovery threshold in terms of the spectral properties of the estimator by a linear stability argument [53]

**Theorem 2.3.** *For $\alpha > \alpha_c$, $\gamma_s = \alpha/\alpha_c$ is the largest value of $\gamma$ such that the state evolution 2.2 has a stable fixed point $(\boldsymbol{M}, \boldsymbol{Q})$ with $\boldsymbol{M} \neq \boldsymbol{0}$, $\boldsymbol{Q} \in \mathbb{S}_+^p \setminus \{\boldsymbol{0}\}$. Additionally, $\boldsymbol{M} \in \mathbb{S}_+^p \setminus \{\boldsymbol{0}\}$.*

**Theorem 2.4.** *For all $\alpha \in \mathbb{R}$, $\gamma_b = \sqrt{\alpha/\alpha_c}$ is the largest value for $\gamma$ such that the state evolution 2.2 has a fixed point $(\boldsymbol{M}, \boldsymbol{Q})$ with $\boldsymbol{M} = \boldsymbol{0}$, $\boldsymbol{Q} \in \mathbb{S}_+^p \setminus \{\boldsymbol{0}\}$. The fixed point is stable for $\alpha < \alpha_c$ and unstable otherwise.*

The derivation of the Theorems is outlined in Appendix B, where we further show that the iterations of Algorithm 2.1 correspond to the power-iteration on the operator $\boldsymbol{L}$, normalized by $\gamma$. Assuming the convergence of Algorithm 2.1 in $\mathcal{O}(\log d)$ iterations, Theorem 2.3 thus implies that for $\alpha > \alpha_c$ the top-most eigenvector of $\boldsymbol{L}$ achieves a non-vanishing overlap along $\boldsymbol{W}_\star$, with the eigenvalues converging to $\gamma_s$. Analogously, Theorem 2.4 implies that for $\alpha < \alpha_c$, the top-most eigenvector of $\boldsymbol{L}$ has a vanishing overlap along $\boldsymbol{W}_\star$, with the eigenvalues converging to $\gamma_b$.

Thus, the above two results indicate a change of behavior of the operator norm of $\boldsymbol{L}$ at the critical value $\alpha_c$, leading to the following conjecture (a detailed sketch can be found in Appendix D):

**Conjecture 2.5.** *In the high-dimensional limit $n, d \to \infty$, $n/d \to \alpha$, the empirical spectral distribution associated to the $pn$ eigenvalues of $\boldsymbol{L}$ converges weakly almost surely to a density whose support is strictly contained in a disk of radius $\gamma_b = \sqrt{\alpha/\alpha_c}$ centered at the origin. Moreover*

- *for $\alpha < \alpha_c$, $\|\boldsymbol{L}\|_{\mathrm{op}} \xrightarrow[n\to\infty]{a.s.} \gamma_b$ and the associated eigenvector is not correlated with $\boldsymbol{W}_\star$;*

- *for $\alpha > \alpha_c$, $\|\boldsymbol{L}\|_{\mathrm{op}} \xrightarrow[n\to\infty]{a.s.} \gamma_s > \gamma_b$ and the associated eigenvector defined in (10), weakly recovers the signal.*

This conjecture, motivated by the results (2.3-2.4) is perfectly supported by extensive simulations, as illustrated in Fig. 1, 2. An entire rigorous proof requires, however, a fine control of the spectral norm of these operators, which is a notably difficult problem in random matrix theory. We emphasize that the asymptotic characterization of the asymmetric spectral estimator is a novelty aspect of this work, even in the single-index model setting $p = 1$.

## 2.2 Symmetric spectral method

In order to simplify the notation, define the symmetric matrices $\boldsymbol{\mathcal{T}}(y) \in \mathbb{R}^{p \times p}$ and $\hat{\boldsymbol{G}}_{\boldsymbol{\mathcal{T}}}^t \in \mathbb{R}^{np \times np}$

$$\boldsymbol{\mathcal{T}}(y) := \boldsymbol{G}(y) \left(\boldsymbol{G}(y) + \boldsymbol{I}_p\right)^{-1} = \boldsymbol{I}_p - \mathrm{Cov}^{-1}[\boldsymbol{z}|y]. \tag{22}$$

$$\left[\hat{\boldsymbol{G}}_{\boldsymbol{\mathcal{T}}}^t\right]_{(i\mu),(j\nu)} := \delta_{ij}[\boldsymbol{\mathcal{T}}(\mathrm{y}_i)\boldsymbol{V}_t \left(a_t \boldsymbol{I}_p - \boldsymbol{V}_t \boldsymbol{\mathcal{T}}(\mathrm{y}_i)\boldsymbol{V}_t\right)^{-1}]_{\mu\nu}, \quad i, j \in [\![n]\!],\ \mu\nu \in [\![p]\!]. \tag{23}$$

**Definition 2.6.** Consider the linear GAMP algorithm (40,41)

$$\boldsymbol{\Omega}^t = \mathbf{X}\hat{\mathbf{W}}^t - \mathrm{mat}\left(\hat{\boldsymbol{G}}_{\boldsymbol{\mathcal{T}}}^{t-1} \mathrm{vec}\left(\boldsymbol{\Omega}^{t-1}\right)\right)\boldsymbol{V}_t, \tag{24}$$

$$\hat{\mathbf{W}}^{t+1} = \left(\mathbf{X}^T \mathrm{mat}\left(\hat{\boldsymbol{G}}_{\boldsymbol{\mathcal{T}}}^t \mathrm{vec}\left(\boldsymbol{\Omega}^t\right)\right) - \hat{\mathbf{W}}^t \boldsymbol{A}_t\right)\boldsymbol{V}_{t+1} \tag{25}$$

with

$$\boldsymbol{V}_{t+1} = \left(\gamma_t \boldsymbol{V}_t - \alpha \mathbb{E}_{\mathrm{y}\sim\mathsf{Z}}\left[\boldsymbol{\mathcal{T}}(\mathrm{y})\boldsymbol{V}_t \left(a_t \boldsymbol{I}_p - \boldsymbol{V}_t \boldsymbol{\mathcal{T}}(\mathrm{y})\boldsymbol{V}_t\right)^{-1}\right]\right)^{-1}, \tag{26}$$

$$[\boldsymbol{A}_t]_{\mu\nu} = d^{-1}\sum_{i\in[\![n]\!]}\left[\hat{\boldsymbol{G}}_{\boldsymbol{\mathcal{T}}}\right]_{(i\mu),(i\nu)}, \quad \mu, \nu \in [\![p]\!], \tag{27}$$

$a_t$ a parameter to be fixed and $\gamma_t$ chosen, a posteriori, such that $\|\boldsymbol{V}\|_{\mathrm{op}} = 1$. Note that $\boldsymbol{V}_t$ is symmetric at all times given a symmetric initialization $\boldsymbol{V}_0$.

In Appendix C we show that, for properly chosen $a_t$, the fixed point for the iterate $\hat{\mathbf{W}}^t \boldsymbol{V}_t$ is the eigenvector of $\boldsymbol{T}$ with eigenvalue $\lim_{t\to\infty} a_t\gamma_t$. The denoiser functions chosen for this GAMP algorithm are derived as a generalization of the ones in [54], where a similar approach has been used to characterize the recovery properties of spectral algorithms for structured single-index models.

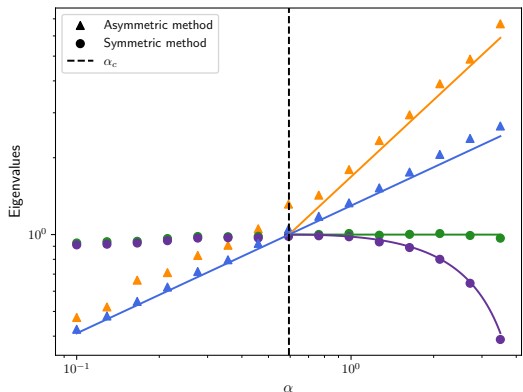

**Figure 2:** Largest eigenvalues (magnitude, if complex) of the matrices $\boldsymbol{L}$ (triangles), $n = 5000$, and $\boldsymbol{T}$ (circles), $d = 5000$, for the function $g(z_1, z_2) = z_1 z_2$, versus the sample complexity $\alpha$. The orange and blue lines, respectively represent the values of the largest eigenvector $\alpha/\alpha_c$ and the edge of the bulk $\sqrt{\alpha/\alpha_c}$ in Conj. 2.5 for the asymmetric spectral method. The green and purple line correspond to the values of the largest eigenvalue $\lambda_s$ and the edge of the bulk $\lambda_b$ in Conj. 2.10 for the symmetric spectral method.

**Proposition 2.7** (State evolution [52, 9]). *Let $\boldsymbol{M}^t$ and $\boldsymbol{Q}^t$ denote the overlaps defined in eq. (16) for the iterative algorithm (2.6). Then, in the proportional high-dimensional limit $n, d \to \infty$ at fixed $\alpha = n/d$, they satisfy the following* state evolution *equations:*

$$\boldsymbol{M}^{t+1} = \alpha \mathcal{F}(\boldsymbol{M}^t; a_t, \boldsymbol{V}_t), \tag{28}$$

$$\boldsymbol{Q}^{t+1} = \boldsymbol{M}^t (\boldsymbol{M}^t)^T + \alpha(\mathcal{G}(\boldsymbol{M}^t; a_t, \boldsymbol{V}_t) + \tilde{\mathcal{F}}(\boldsymbol{Q}^t; a_t, \boldsymbol{V}_t)), \tag{29}$$

*where we have defined the operators*

$$\mathcal{F}(\boldsymbol{M}; a_t, \boldsymbol{V}_t) := \mathbb{E}_{y \sim Z} \left[ \boldsymbol{C}_t \, \boldsymbol{M} \left( \mathrm{Cov}[\mathbf{z}|y] - \boldsymbol{I}_p \right) \right], \tag{30}$$

$$\tilde{\mathcal{F}}(\boldsymbol{Q}; a, \boldsymbol{V}) := \mathbb{E}_{y \sim Z} \left[ \boldsymbol{C}_t \, \boldsymbol{Q} \, \boldsymbol{C}_t \right] \tag{31}$$

$$\mathcal{G}(\boldsymbol{M}; a, \boldsymbol{V}) := \mathbb{E}_{y \sim Z} \left[ \boldsymbol{C}_t \, \boldsymbol{M} \left( \mathrm{Cov}[\mathbf{z}|y] - \boldsymbol{I}_p \right) \boldsymbol{M}^T \, \boldsymbol{C}_t \right]. \tag{32}$$

*with $\boldsymbol{C}_t := \boldsymbol{V}_t \boldsymbol{\mathcal{T}}(y) \boldsymbol{V}_t \left( a_t - \boldsymbol{V}_t \boldsymbol{\mathcal{T}}(y) \boldsymbol{V}_t \right)^{-1}$.*

The complete set of state evolution equations is displayed in Appendix C.1. For $a = 1$, the fixed point for eq. (26) is $\boldsymbol{V} = \boldsymbol{I}_p$ and the operators $\mathcal{F}(\boldsymbol{M}; 1, \boldsymbol{I}_p)$ and $\tilde{\mathcal{F}}(\boldsymbol{M}; 1, \boldsymbol{I}_p)$ coincide with $\mathcal{F}(\boldsymbol{M})$ defined in (8), having the largest eigenvalue corresponding to $\alpha_c^{-1}$. As $a \to \infty$, $\boldsymbol{V} \to \boldsymbol{I}_p$ and $\mathcal{F}(\,\cdot\,; a, \boldsymbol{V}(a)) \to \boldsymbol{0}$. Therefore, since the operator depends continuously on $a \in [1, \infty)$, there exists a continuous function $\nu_1^{\mathcal{F}}(a)$ for the largest eigenvalue of the operator as a function of the parameter $a$, with $\nu_1^{\mathcal{F}}(1) = \alpha_c^{-1}$ and $\nu_1^{\mathcal{F}}(a \to \infty) \to 0$. Hence, for all $\alpha > \alpha_c$, there exists $a > 1$ such that $\nu_1^{\mathcal{F}}(a) = \alpha^{-1}$. A similar argument applies to $\tilde{\mathcal{F}}$ and its largest eigenvalue $\nu_1^{\tilde{\mathcal{F}}}(a)$.

**Theorem 2.8.** *For $\alpha > \alpha_c$, consider $a_t = a > 1$ solution of $\nu_1^{\mathcal{F}}(a_t) = \alpha^{-1}$ and $\gamma_t$ such that $\|\boldsymbol{V}\|_{\mathrm{op}} = 1$, with $\boldsymbol{V}_{t+1}$ given by eq. (26). Then, the state evolution has a stable fixed point $(\boldsymbol{M}, \boldsymbol{Q})$ with $\boldsymbol{M} \neq \boldsymbol{0}$, $\boldsymbol{Q} \in \mathbb{S}_+^p \setminus \{\boldsymbol{0}\}$, corresponding to the eigenvalue $\lambda_s = a\gamma$.*

**Theorem 2.9.** *For $\alpha \geq \alpha_c$, $a_t = a \geq 1$ solution to $\nu_1^{\tilde{\mathcal{F}}}(a) = \alpha^{-1}$, $\gamma_t$ such that $\|\boldsymbol{V}\|_{\mathrm{op}} = 1$, with $\boldsymbol{V}_{t+1}$ given by eq. (26), the state evolution has an unstable fixed point $(\boldsymbol{M}, \boldsymbol{Q})$ with $\boldsymbol{M} = \boldsymbol{0}$, $\boldsymbol{Q} \in \mathbb{S}_+^p \setminus \{\boldsymbol{0}\}$, corresponding to the eigenvalue $\lambda_b = a\gamma$.*

The derivation of the Theorems is outlined in Appendix C. Analogously to how Theorems 2.3, 2.4 motivate Conjecture 2.5, we show in Appendix C that Theorems 2.8, 2.9, along with a mapping to power-iteration, lead to the following conjecture:

**Conjecture 2.10.** *In the high-dimensional limit $n, d \to \infty$, $n/d \to \alpha$, for $\alpha > \alpha_c$, the largest eigenvalue of $\boldsymbol{T}$ converges to $\lambda_s$, defined in Theorem 2.8. In this regime, the symmetric spectral method, defined in (12), weakly recovers the signal. Moreover, the empirical spectral distribution of the $pd$ eigenvalues of $\boldsymbol{T}$ converges weakly almost surely to a density upper bounded by $\lambda_b < \lambda_s$ defined in Theorem 2.9.*

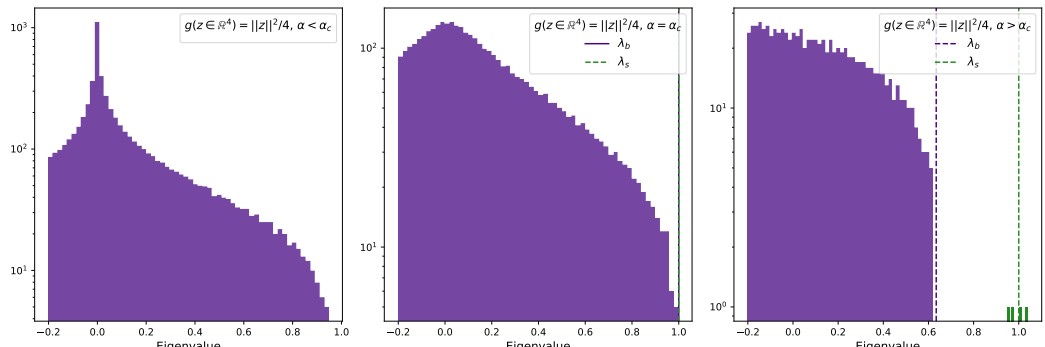

**Figure 3:** Distribution of the eigenvalues of $\boldsymbol{T}$, $d = 10^4$, for the link function $g(\boldsymbol{z}) = p^{-1}\|\boldsymbol{z}\|^2$, $p = 4$. The critical threshold in $\alpha_c = 2$. The distribution is truncated on the left. (**Left**) $\alpha = 1 < \alpha_c$. (**Center**) $\alpha = \alpha_c$. (**Right**) $\alpha = 6 > \alpha_c$. As predicted by the state evolution, we observe four eigenvalues (in green) separated from the main bulk, centered around $\lambda_s = 1$ (green vertical line) obtained in Theorem 2.8. The vertical purple line correspond to the value $\lambda_b$ provided in Theorem 2.9 as a bound for the bulk.

We emphasize that, when $p = 1$, the proposed symmetric estimator specializes to the method in [9, 55, 12] for the single-index model, and therefore benefits from a fully rigorous characterization of its weak recovery properties and spectral phase transitions. However, in multi-index models ($p > 1$), the matrix $\boldsymbol{T}$ exhibits by construction a highly structured form due to the presence of *repeated* entries from the measurement matrix $\mathbf{X}$. This intrinsic redundancy complicates its analysis using standard random matrix theory tools. Conjecture 2.10 based on results (2.8-2.9) and further supported by numerical simulations (see Fig. 2, 3), offers a novel framework for understanding the spectral properties of such matrices.

Moreover, for any model $\mathsf{P}(\cdot|\boldsymbol{z})$ such that $\boldsymbol{G}(y) = \mathbb{E}[\mathbf{z}\mathbf{z}^T - \boldsymbol{I}_p|\mathrm{y} = y]$ admits a common basis for all $y$, with real eigenvalues $\{\lambda_k(y)\}_{k=1}^p$, the analysis of spectrum of $\boldsymbol{T}$ can be simplified following the arguments in Appendix C.3.1. Indeed, the symmetric spectral method reduces to the diagonalization of $p$ matrices

$$\sum_{i \in [\![n]\!]} \frac{\lambda_k(\mathrm{y}_i)}{\lambda_k(\mathrm{y}_i) + 1} \mathbf{x}_i \mathbf{x}_i^T \in \mathbb{R}^{d \times d}, \quad k \in [\![p]\!]. \tag{33}$$

Their structure allows the use of the techniques in [9, 55] to analyze the spectrum, and supports the formalization of the results in Conjecture 2.10 for this subset of problems. In Appendix C.3 we prove the following result

**Theorem 2.11.** *Assume that the matrix $\mathbb{E}[\mathbf{z}\mathbf{z}^T|y]$ admits a basis of orthonormal eigenvectors independent of $y$. Then, in the high-dimensional limit $n, d \to \infty$, $n/d \to \alpha$, for $\alpha > \alpha_c$, the largest eigenvalue of $\boldsymbol{T}$ converges to $\lambda_s = 1$. Moreover, the empirical spectral distribution of the $pd$ eigenvalues of $\boldsymbol{T}$ converges weakly almost surely to a density upper bounded by $\lambda_b < 1$ defined in Theorem 2.9.*

This class of models $\mathsf{P}(\cdot|\boldsymbol{z})$ includes several examples of interest, such as those depending on $\boldsymbol{z}$ through a quadratic form $\boldsymbol{z}^\top \boldsymbol{A} \boldsymbol{z}$ with deterministic $\boldsymbol{A}$ (e.g., the norm $\|\boldsymbol{z}\|^2$), or through the product $z_1 z_2 \ldots z_p$ (e.g., the embedded sparse parity).

### 2.3 Relation between the two spectral methods

As in the single-index setting, $\boldsymbol{L}$ and $\boldsymbol{T}$ are related by the following Proposition (proven in App. G).

**Proposition 2.12.** *Define $\hat{\boldsymbol{G}} \in \mathbb{R}^{np \times np}$ such that $\hat{G}_{(i\mu),(j\nu)} := \delta_{ij} G_{\mu\nu}(\mathrm{y}_i)$.*

*1. Given an eigenpair $\gamma_{\boldsymbol{L}} \geq 1, \boldsymbol{\omega} \in \mathbb{R}^{np}$ of $\boldsymbol{L}$, if $\boldsymbol{w} := \mathrm{vec}\left(\mathbf{X}^T \mathrm{mat}\left(\hat{\boldsymbol{G}}\boldsymbol{\omega}\right)\right)$, then*

$$\boldsymbol{T}_{\gamma_{\boldsymbol{L}}} \boldsymbol{w} = \boldsymbol{w}, \tag{34}$$

*with $\boldsymbol{T}_\gamma \in \mathbb{R}^{dp \times dp}$ defined as*

$$[\boldsymbol{T}_\gamma]_{(k\mu),(h\nu)} := \sum_{i \in [\![n]\!]} \mathrm{X}_{ik}\mathrm{X}_{ih}\left[\boldsymbol{G}(\mathrm{y}_i)\left(\boldsymbol{G}(\mathrm{y}_i) + \gamma\boldsymbol{I}_p\right)^{-1}\right]_{\mu\nu} \quad k, h \in [\![d]\!], \, \mu, \nu \in [\![p]\!], \tag{35}$$

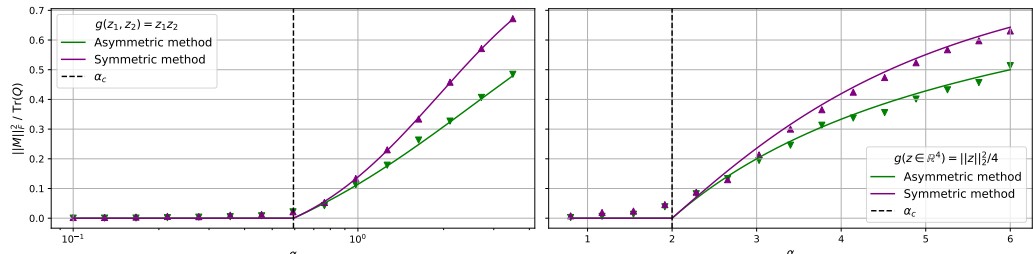

**Figure 4:** Overlap $\|M\|_{\mathrm{F}}^2/\mathrm{Tr}(Q)$ as a function of the sample complexity $\alpha$. The dots represent numerical simulation results, computed for $n = 5000$ (for the asymmetric method) or $d = 5000$ (for the symmetric method) and averaging over 10 instances. (**Left**) Link function $g(z_1, z_2) = z_1 z_2$. Solid lines are obtained from state evolution predictions (Section 3). Dashed line at $\alpha_c \approx 0.59375$. (**Right**) Link function $g(z) = p^{-1}\|z\|^2$, $p = 4$. Solid lines are obtained from state evolution predictions (Section 3). Dashed line at $\alpha_c = 2$.

2. *Given an eigenpair* $\gamma_T, w \in \mathbb{R}^{dp}$ *of* $T$ *and defining* $\omega := (I_{np} + \hat{G})^{-1}\mathrm{vec}\left(X\mathrm{mat}(w)\right)$, *then*

$$L\omega = \gamma_T \omega + (\gamma_T - 1)\hat{G}\omega. \tag{36}$$

*Consequently, if there exists an eigenvector* $w$ *of* $T$ *with eigenvalue* $\gamma_T = 1$, *then* $\omega = (I_{np} + \hat{G})^{-1}\mathrm{vec}\left(X\mathrm{mat}(w)\right)$ *is an eigenvector of* $L$ *with eigenvalue* $\gamma_L = \gamma_T = 1$.

In the setting of Theorem 2.11, where the largest eigenvalue of $T$ is $\lambda_s = 1$, Proposition 2.12 implies that the principal eigenvector of $L$ is not the only one correlated with the signal. There exists at least one additional informative eigenvector, whose eigenvalue is one and lies hidden within the bulk of radius $\sqrt{\alpha/\alpha_c} > 1$. This phenomenon is reminiscent of what has been observed for single-index models in [10].

## 3 Numerical illustrations

In this section we illustrate the framework introduced in Section 2 to predict the asymptotic performance of the spectral estimators (10,12) for specific examples of link functions, providing a comparison between our asymptotic analytical results and finite size numerical simulations for the overlap between the spectral estimators and the weights $W_\star$, defined as $m := \|M\|_{\mathrm{F}}/\sqrt{\mathrm{Tr}(Q)}$, where $M$ and $Q$ are the overlap matrices defined in eq. (16) correspondent to the fixed points in Theorems 2.3, 2.4, 2.8, 2.9. In Figure 4 we compare these theoretical predictions to numerical simulations at finite dimensions, respectively for the link functions $g(z_1, z_2) = z_1 z_2$ and $g(z) = p^{-1}\|z\|^2$. Additional numerical experiments are presented in Appendix E.

**Asymmetric spectral method.** We provide closed-form expressions for the overlap parameter $m := \|M\|_{\mathrm{F}}/\sqrt{\mathrm{Tr}(Q)}$ of the spectral estimator $\hat{W}_L$ (10), for a selection of examples of link functions. The details of the derivation are given in Appendix E.

- $g(z \in \mathbb{R})$ (single-index model):

$$\alpha_c = \left(\mathbb{E}_{y \sim Z}\left[\left(\mathbb{E}[z^2 - 1|y]\right)^2\right]\right)^{-1}, \quad m^2 = \left(\frac{\alpha - \alpha_c}{\alpha + \alpha_c^2 \mathbb{E}_{y \sim Z}\left[\left(\mathbb{E}[z^2 - 1|y]\right)^3\right]}\right)_+$$

- $g(z) = p^{-1}\|z\|^2$:

$$\alpha_c = p/2, \quad m^2 = \left((\alpha - p/2)(\alpha + 2)^{-1}\right)_+$$

- $g(z) = \mathrm{sign}(z_1 z_2)$:

$$\alpha_c = \pi^2/4, \quad m^2 = \left(1 - \frac{\pi^2}{4\alpha}\right)_+$$

- $g(\boldsymbol{z}) = \prod_{k=1}^{p} z_k$:

$$\alpha_c = \left(\mathbb{E}_{\mathrm{y}\sim\mathsf{Z}}\left[\lambda(\mathrm{y})^2\right]\right)^{-1}, \quad m^2 = \left(\frac{\alpha - \alpha_c}{\alpha + \alpha_c^2 \mathbb{E}_{\mathrm{y}\sim\mathsf{Z}}\left[\lambda(\mathrm{y})^3\right]}\right)_+,$$

  where we have used

$$\lambda(y) = |y|\frac{K_1(|y|)}{K_0(|y|)} + \mathrm{y} - 1, \quad (p=2) \quad \text{and} \quad \lambda(y) = \frac{2G_{0,p}^{p,0}\left(y^2 2^{-p} \,\middle|\, \begin{matrix} 0 \\ \boldsymbol{e}_p \end{matrix}\right)}{G_{0,p}^{p,0}\left(y^2 2^{-p} \,\middle|\, \begin{matrix} 0 \\ \boldsymbol{0}_p \end{matrix}\right)} - 1, \quad (p \geq 3)$$

(37)

  and the previous expression are written in terms of the modified Bessel function of the second kind and Meijer $G$-function, with the notations $\boldsymbol{0}_p \in \mathbb{R}^p = (0, \ldots, 0)^T$ and $\boldsymbol{e}_p \in \mathbb{R}^p = (0, \ldots, 0, 1)^T$.

- $g(z_1, z_2) = z_1 z_2^{-1}$:

$$\alpha_c = 1, \quad m^2 = (1 - \alpha^{-1})_+$$

**Symmetric spectral method.** We provide expressions for the overlap parameter $m := \|\boldsymbol{M}\|_{\mathrm{F}}/\sqrt{\mathrm{Tr}(\boldsymbol{Q})}$ of the spectral estimator $\hat{\mathbf{W}}_{\boldsymbol{T}}$ (12), for a selection of examples of link functions. In all the following cases, the state evolution equations simplify, allowing to write the results as functionals of $\lambda : \mathbb{R} \to \mathbb{R}$, specific to each problem:

- $g(z \in \mathbb{R})$ (single-index model): $\lambda(y) = \mathrm{Var}[z \mid y] - 1$;
- $g(\boldsymbol{z}) = \mathrm{sign}(z_1 z_2)$: $\lambda(y) = 2y/\pi$
- $g(\boldsymbol{z}) = p^{-1}\|\boldsymbol{z}\|^2$: $\lambda(y) = y - 1$;
- $g(\boldsymbol{z}) = \prod_{k=1}^{p} z_k$: $\lambda(y)$ defined in eq. (37).

For all these examples, the value $\alpha_c$ has been reported in the previous paragraph. For $\alpha > \alpha_c$, consider $a$ and $\gamma$ solutions of

$$\mathbb{E}_{\mathrm{y}\sim\mathsf{Z}}\left[\frac{\lambda(\mathrm{y})^2}{a(1 + \lambda(\mathrm{y})) - \lambda(\mathrm{y})}\right] = \frac{1}{\alpha}, \qquad \gamma = 1 + \alpha\mathbb{E}_{\mathrm{y}\sim\mathsf{Z}}\left[\frac{\lambda(\mathrm{y})}{a(1 + \lambda(\mathrm{y})) - \lambda(\mathrm{y})}\right]. \tag{38}$$

Then, for any $\alpha$, the overlap $m := \|\boldsymbol{M}\|_{\mathrm{F}}/\sqrt{\mathrm{Tr}(\boldsymbol{Q})}$ is given by

$$m^2 = \left(\frac{1 - \alpha\mathbb{E}_{\mathrm{y}\sim\mathsf{Z}}[\lambda^2(\mathrm{y})\left(a(1 + \lambda(\mathrm{y})) - \lambda(\mathrm{y})\right)^{-2}]}{1 + \alpha\mathbb{E}_{\mathrm{y}\sim\mathsf{Z}}[\lambda^3(\mathrm{y})\left(a(1 + \lambda(\mathrm{y})) - \lambda(\mathrm{y})\right)^{-2}]}\right)_+, \tag{39}$$

which is strictly positive $\forall \alpha > \alpha_c$. In all the example the principal eigenvector is $\lambda_s = 1$. Additional details can be found in Appendix F.

## 4 Conclusion and Perspectives

In this work, we tackled weak recovery in high-dimensional multi-index models via spectral methods, deriving two estimators inspired by a linearization of AMP. We showed they achieve the optimal reconstruction threshold, closing a key gap in prior approaches that required additional side information. Our analysis establishes that above the critical sample complexity, the leading eigenvectors of the proposed spectral operators align with the ground-truth subspace, echoing the BBP transition in random matrix theory. This work advances our understanding of weak subspace recovery in multi-index models and provides a principled framework for designing optimal spectral estimators. It bridges ideas from random matrix theory, approximate message passing, and neural feature learning.

Several directions remain open. A random matrix theory analysis of our spectral analysis — which requires a challenging control of the spectral norms — could be used to prove the two conjectures. Extending our AMP linearization to higher-order schemes, such as the Kikuchi hierarchy, may unlock insights into harder generative exponent problems, including the notorious sparse parity function. We hope this work sparks further research at the intersection of spectral methods and high-dimensional inference.

## Acknowledgements

The authors would like to thank Antoine Maillard, Benjamin Aubin and Lenka Zdeborova for early discussions on this problems in KITP Santa Barbara. We would also like to thank Filip Kovačević, Yihan Zhang and Marco Mondelli for the discussions on the relationship with their work following the first version of this manuscript. This research was supported in part by grant NSF PHY-2309135 to the Kavli Institute for Theoretical Physics (KITP), by the Swiss National Science Foundation under grant SNSF OperaGOST (grant number 200390) and DGIANGO (grant number 225837), and by the French government, managed by the National Research Agency (ANR), under the France 2030 program with the reference "ANR-23-IACL-0008" and the Choose France - CNRS AI Rising Talents program.

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

## A Generalized Approximate Message Passing algorithms

In this section we present a general version of the multi-dimensional Generalized Approximate Message Passing (GAMP) algorithm [32], defined as the iterations

$$\boldsymbol{\Omega}^t = \mathbf{X}\boldsymbol{f}_{in}^t(\boldsymbol{B}^t) - \boldsymbol{g}_{out}^{t-1}(\boldsymbol{\Omega}^{t-1}, \boldsymbol{y})\boldsymbol{V}_t^T, \tag{40}$$

$$\boldsymbol{B}^{t+1} = \mathbf{X}^T\boldsymbol{g}_{out}^t(\boldsymbol{\Omega}^t, \boldsymbol{y}) - \boldsymbol{f}_{in}^t(\boldsymbol{B}^t)\boldsymbol{A}_t^T \tag{41}$$

and $\hat{\mathbf{W}}^{t+1} = \boldsymbol{f}_{in}^{t+1}(\boldsymbol{B}^{t+1})$. The *denoiser* functions $\boldsymbol{f}_{in}^t : \mathbb{R}^p \to \mathbb{R}^p$ and $\boldsymbol{g}_{out}^t : \mathbb{R}^p \times \mathbb{R} \to \mathbb{R}^p$ are vector-valued mappings acting row-wise respectively on $\boldsymbol{b}_j \in \mathbb{R}^p = \boldsymbol{B}_{j.}$ and $\boldsymbol{\omega}_i \in \mathbb{R}^p = \boldsymbol{\Omega}_{i.}$ and the *Onsager terms* are given by

$$\boldsymbol{A}_t = \frac{1}{d}\sum_{i=1}^n \nabla_{\boldsymbol{\omega}_i}\boldsymbol{g}_{out}^t(\boldsymbol{\omega}_i, y_i), \qquad \boldsymbol{V}_t = \frac{1}{d}\sum_{j=1}^d \nabla_{\boldsymbol{b}_j}\boldsymbol{f}_{in}^t(\boldsymbol{b}_j). \tag{42}$$

Therefore, the algorithm is uniquely determined by the choice of denoisers. For instance, the optimal GAMP for Gaussian multi-index models, derived in [4], is given by

$$\boldsymbol{g}_{out}^t(\boldsymbol{\omega}, y) = \boldsymbol{V}_t^{-1}\mathbb{E}_{\mathbf{z}\sim\mathsf{N}(\boldsymbol{\omega}, \boldsymbol{V}_t)}[(\mathbf{z}-\boldsymbol{\omega})\mathsf{P}(y|\mathbf{z})], \qquad \boldsymbol{f}_{in}^t(\boldsymbol{b}) = \left(\boldsymbol{I}_p - \frac{1}{d}\sum_{i=1}^n \nabla_{\boldsymbol{\omega}_i}\boldsymbol{g}_{out}^t(\boldsymbol{\omega}_i^t, y_i)\right)^{-1}\boldsymbol{b}, \tag{43}$$

### A.1 Linear GAMP

In this manuscript we focus on a special type of GAMP algorithms that have linear denoiser functions

$$\boldsymbol{g}_{out}^t(\boldsymbol{\omega}, y) = \boldsymbol{G}_{out}^t(y)\boldsymbol{\omega}, \quad \boldsymbol{f}_{in}^t(\boldsymbol{b}) = \boldsymbol{V}^t\boldsymbol{b}, \tag{44}$$

namely

$$\boldsymbol{\Omega}^t = \mathbf{X}\hat{\mathbf{W}}^t - \boldsymbol{\Gamma}_{out}^{t-1}\boldsymbol{V}_t^T, \tag{45}$$

$$[\boldsymbol{\Gamma}_{out}^t]_{i\mu} = \sum_{\nu\in[\![p]\!]} [\boldsymbol{G}_{out}^t]_{\mu\nu}(y_i)\Omega_{i\nu}^t, \quad i\in[\![n]\!], \mu\in[\![p]\!], \tag{46}$$

$$\hat{\mathbf{W}}^{t+1} = \left(\mathbf{X}^T\boldsymbol{\Gamma}_{out}^t - \hat{\mathbf{W}}^t\boldsymbol{A}_t^T\right)\boldsymbol{V}_{t+1}^T, \tag{47}$$

where

$$\boldsymbol{A} = d^{-1}\sum_{i\in[\![n]\!]}\boldsymbol{G}_{out}(y_i) \xrightarrow{n,d\to\infty} \alpha\mathbb{E}_{y\sim\mathsf{Z}}[\boldsymbol{G}_{out}(y)]. \tag{48}$$

A particular example of Linear GAMP is the one obtained linearizing the denoiser functions (43) around the uninformed fixed point of the algorithm $\boldsymbol{b} = 0$, $\boldsymbol{\omega} = \boldsymbol{0}$ and $\boldsymbol{V} = \boldsymbol{I}_p$. We obtain a Linear GAMP with

$$\boldsymbol{G}_{out}(y) = \mathbb{E}[\mathbf{z}\mathbf{z}^T - \boldsymbol{I}|\mathbf{y} = y], \quad \boldsymbol{V} = \boldsymbol{I} \tag{49}$$

and

$$\boldsymbol{A} = \alpha\mathbb{E}_{y\sim\mathsf{Z}}[\boldsymbol{G}_{out}(y)] = \alpha\mathbb{E}_{y\sim\mathsf{Z}}\mathbb{E}[\mathbf{z}\mathbf{z}^T|\mathbf{y}] - \alpha\boldsymbol{I} = \alpha\mathbb{E}_{\mathbf{z}\sim\mathsf{N}(\boldsymbol{0},\boldsymbol{I})}[\mathbf{z}\mathbf{z}^T] - \alpha\boldsymbol{I} = \boldsymbol{0}. \tag{50}$$

### A.2 State Evolution of Linear GAMP

One of the main advantages provided by the Approximate Message Passing algorithm is the possibility to track the value of low-dimensional functions of the iterates at all finite times, in the high-dimensional limit, through a set of iterative equations denoted as *state evolution* [52]. In particular, we are interested to the following overlap matrices

$$\boldsymbol{M}^t := \frac{1}{d}\left(\hat{\mathbf{W}}^t\right)^T\boldsymbol{W}_\star, \quad \boldsymbol{Q}^t := \frac{1}{d}\left(\hat{\mathbf{W}}^t\right)^T\hat{\mathbf{W}}^t, \tag{51}$$

that characterize respectively the alignment between $\hat{\mathbf{W}}^t$ and the weights $\boldsymbol{W}_\star$ and the norm of $\hat{\mathbf{W}}^t$. In this appendix we present the state evolution equations for the GAMP algorithm (45, 47) with linear denoiser functions, while we refer to [4] for a complete derivation in more general settings:

$$\boldsymbol{M}^{t+1} = \boldsymbol{V}_t\hat{\boldsymbol{M}}^t, \qquad \boldsymbol{Q}^{t+1} = \boldsymbol{V}_t\left(\hat{\boldsymbol{M}}^t(\hat{\boldsymbol{M}}^t)^T + \hat{\boldsymbol{Q}}^t\right)\boldsymbol{V}_t^T, \tag{52}$$

with the auxiliary matrices given by

$$\hat{M}^t = \alpha \mathbb{E}_{y \sim Z} \left[ G_{\text{out}}(y) M^t \mathbb{E}[\mathbf{z}\mathbf{z}^T - I|y] \right], \tag{53}$$

$$\hat{Q}^t = \alpha \left( \mathbb{E}_{y \sim Z} \left[ G_{\text{out}}(y) M^t \mathbb{E}[\mathbf{z}\mathbf{z}^T - I|y] M^T G_{\text{out}}(y)^T \right] + \mathbb{E}_{y \sim Z} \left[ G_{\text{out}}(y) Q^t G_{\text{out}}(y)^T \right] \right). \tag{54}$$

# B   Proof Outlines for Theorems 2.3 and 2.4 – Asymmetric Spectral Method

Consider the following generalized power iteration algorithm

$$\boldsymbol{\omega}^t = \gamma^{-1} \boldsymbol{L} \, \boldsymbol{\omega}^{t-1} \in \mathbb{R}^{np}, \tag{55}$$

$$\hat{\mathbf{W}}^{t+1} = \gamma^{-1} \mathbf{X}^T \text{mat} \left( \hat{G} \, \boldsymbol{\omega}^t \right) \in \mathbb{R}^{d \times p}, \tag{56}$$

with $\hat{G}$ defined in eq. (9) and $\gamma$ a parameter to be fixed. A pair $\boldsymbol{\omega} \neq \mathbf{0}$, $\hat{\mathbf{W}} = \gamma^{-1} \mathbf{X}^T \text{mat} \left( \hat{G} \boldsymbol{\omega} \right)$, is a fixed point of the above algorithm if and only if $\boldsymbol{L}\boldsymbol{\omega} = \gamma \boldsymbol{\Omega}$. Thus, given the largest real $\gamma$ such that the algorithm has a non-zero fixed point, the latter will correspond to the asymmetric spectral estimator in Definition 10. Interestingly, the above algorithm is also linear GAMP A.1, with denoiser functions $\boldsymbol{g}_{\text{out}}^t(\boldsymbol{\omega} \in \mathbb{R}^p, y) = \mathbb{E}[\mathbf{z}\mathbf{z}^T - I|y = y]\boldsymbol{\omega}$ and $\boldsymbol{f}_{in}^t(\boldsymbol{b}) = \gamma^{-1}\boldsymbol{b}, \forall t$, and Onsager terms

$$\boldsymbol{A}_t \xrightarrow{n \to \infty} \alpha \mathbb{E}_{y \sim Z} \left[ \mathbb{E}[\mathbf{z}\mathbf{z}^T - I|y] \right] = \mathbf{0} \tag{57}$$

(as shown in (50)), and $\boldsymbol{V}_t = \gamma^{-1} I$. In fact, we can rewrite the generalized power iteration algorithm as in Definition 2.1

$$\boldsymbol{\Omega}^t = \gamma^{-1}(\mathbf{X}\mathbf{X}^T - \boldsymbol{I}_n)\text{mat}\left( \hat{G} \text{vec} \left( \boldsymbol{\Omega}^{t-1} \right) \right) = \mathbf{X}\hat{\mathbf{W}}^t - \gamma^{-1}\text{mat}\left( \hat{G} \text{vec} \left( \boldsymbol{\Omega}^{t-1} \right) \right), \quad \tag{58}$$

$$\boldsymbol{B}^{t+1} = \mathbf{X}^T \text{mat}\left( \hat{G} \text{vec} \left( \boldsymbol{\Omega}^t \right) \right), \tag{59}$$

$$\hat{\mathbf{W}}^{t+1} = \gamma^{-1} \boldsymbol{B}^{t+1}. \tag{60}$$

## B.1   State Evolution

As an Approximate Message Passing algorithm, this algorithm enables to track low-dimensional functions of the iterate $\hat{\mathbf{W}}^t$ via the associated state evolution. Specifically, we will analyze the weak recovery properties of the asymmetric spectral method by studying the convergence of the state evolution equations A.2:

$$\boldsymbol{M}^{t+1} = \frac{\alpha}{\gamma} \mathcal{F}(\boldsymbol{M}^t) \tag{61}$$

$$\boldsymbol{Q}^{t+1} = \boldsymbol{M}^t(\boldsymbol{M}^t)^T + \frac{\alpha}{\gamma^2} \left( \mathcal{G}(\boldsymbol{M}^t) + \mathcal{F}(\boldsymbol{Q}^t) \right) \tag{62}$$

where, recalling the notation $\boldsymbol{G}(y) = \mathbb{E}[\mathbf{z}\mathbf{z}^T - I|y]$,

$$\mathcal{F}(\boldsymbol{M}) \coloneqq \mathbb{E}_y[\boldsymbol{G}(y)\boldsymbol{M}\boldsymbol{G}(y)], \tag{63}$$

$$\mathcal{G}(\boldsymbol{M}) \coloneqq \mathbb{E}_y[\boldsymbol{G}(y)\boldsymbol{M}\boldsymbol{G}(y)\boldsymbol{M}^T\boldsymbol{G}(y)]. \tag{64}$$

The linear operator $\mathcal{F} : \mathbb{R}^{p \times p} \to \mathbb{R}^{p \times p}$ is symmetric, therefore it admits $p^2$ eigenpairs ($\nu_k \in \mathbb{R}$, $\boldsymbol{M}_k)_{k \in [p^2]}$ such that $\mathcal{F}(\boldsymbol{M}_k) = \nu_k \boldsymbol{M}_k$ and the (matrix) eigenvectors are an orthonormal basis of $\mathbb{R}^{p \times p}$. In particular, Lemma 1.2 implies that $\nu_1 \coloneqq \max_k \nu_k > 0$ and $\boldsymbol{M}_1 \in \mathbb{S}_+^p$. This eigenvalue corresponds to the inverse of the critical sample complexity $\alpha_c$, defined in 1.2. We can distinguish between two kind of fixed points:

1. **Informed fixed points.** (*Theorem 2.3*) $\forall \alpha$, for $\gamma = \alpha \nu_k$ ($\nu_k \neq 0$), $\boldsymbol{M} \propto \boldsymbol{M}_k$, is a non-zero fixed point for eq. (61). In particular, since the asymmetric spectral estimator is the fixed point correspondent to the largest $\gamma$, we are interested to the case $\boldsymbol{M} \propto \boldsymbol{M}_1$. We show now that $\|\boldsymbol{M}\|_F \neq 0$ (*i.e.* the fixed point is actually informative) for $\alpha > \alpha_c$. Given

$\gamma = \alpha/\alpha_c > 1$, the largest eigenvalue of the operator $\gamma^{-1}\alpha\mathcal{F}(\cdot)$ is equal to 1 and any $\boldsymbol{M} \propto \boldsymbol{M}_1$ is a stable fixed point. Moreover, eq. (62) at convergence:

$$\boldsymbol{Q} = \boldsymbol{M}^2 + \frac{\alpha_c^2}{\alpha}(\|\boldsymbol{M}\|_{\mathrm{F}}^2\mathcal{G}(\boldsymbol{M}) + \mathcal{F}(\boldsymbol{Q})) \implies \tag{65}$$

$$\mathrm{Tr}(\boldsymbol{Q}) = \|\boldsymbol{M}\|_{\mathrm{F}}^2 \left( 1 + \frac{\alpha_c^2}{\alpha} \underbrace{\mathrm{Tr}\left( \mathcal{G}\left( \frac{\boldsymbol{M}}{\|\boldsymbol{M}\|_{\mathrm{F}}} \right) \right)}_{\geq 0} \right) + \frac{\alpha_c^2}{\alpha} \underbrace{\mathrm{Tr}(\mathcal{F}(\boldsymbol{Q}))}_{\leq \nu_1 \mathrm{Tr}(\boldsymbol{Q})} \implies \tag{66}$$

$$\|\boldsymbol{M}\|_{\mathrm{F}}^2 \geq \left( 1 - \frac{\alpha_c}{\alpha} \right) \mathrm{Tr}(\boldsymbol{Q}) \left( 1 + \frac{\alpha_c^2}{\alpha}\mathrm{Tr}\left( \mathcal{G}\left( \frac{\boldsymbol{M}}{\|\boldsymbol{M}\|_{\mathrm{F}}} \right) \right) \right)^{-1} > 0, \tag{67}$$

where in eq. (66) we used

$$\mathrm{Tr}\mathcal{F}\left( \boldsymbol{Q} = \sum_{k\in[\![p^2]\!]} c_k\boldsymbol{M}_k \right) = \mathrm{Tr} \sum_{k\in[\![p^2]\!]} c_k\nu_k\boldsymbol{M}_k \leq \nu_1\mathrm{Tr}\boldsymbol{Q}. \tag{68}$$

Therefore, the correspondent estimator $\hat{\mathbf{W}}$ eq. (60) weakly recovers $\boldsymbol{W}_\star$.

Note that, although it is beyond the scope of this work, the presented framework enables the identification of additional informative eigenvectors (with real eigenvalues smaller than $\alpha/\alpha_c$), emerging from the bulk at larger sample complexities.

2. **Uninformed fixed point.** (*Theorem 2.4*) Initializing GAMP in a subspace orthogonal to the signal, *i.e.* $\boldsymbol{M}^0 = \boldsymbol{0}$, we have that $\boldsymbol{M}^t = \boldsymbol{0}$ at all times and eq. (62) at convergence

$$\boldsymbol{Q} = \gamma^{-2}\alpha\mathcal{F}(\boldsymbol{Q}). \tag{69}$$

Therefore, for $\gamma = \sqrt{\alpha\nu_1}$, $\boldsymbol{M} = \boldsymbol{0}$, $\boldsymbol{Q} = \boldsymbol{M}_1 \in \mathbb{S}_+^p \setminus \{\boldsymbol{0}\}$ is a fixed point of the state evolution. Note that, since the largest eigenvalue of $\gamma^{-1}\alpha\mathcal{F}(\cdot)$ in eq. (61) is $\sqrt{\alpha/\alpha_c}$, $\boldsymbol{M} = \boldsymbol{0}$ is a stable fixed point for $\alpha \leq \alpha_c$, and unstable otherwise. Since the proposed GAMP is a generalized power iteration algorithm, normalized by the constant $\gamma \in \mathbb{R}$, it converges only if $\gamma$ corresponds to the absolute value of the eigenvalue with largest magnitude in the subspace of initialization.

## C  Proof Outlines for Theorems 2.8 and 2.9 – Symmetric Spectral Method

Similarly to what we have done in the previous section, we introduce a GAMP algorithm that will serve as a framework to study the properties of the spectral estimator defined in (12). We stress that this algorithm does not offer any particular advantage for the practical computation of the spectral estimator compared to other spectral algorithms.

Consider the Generalized Approximate Message Passing algorithm defined by the denoiser functions

$$\boldsymbol{g}_{\mathrm{out}}^t(y, \boldsymbol{\omega}) = \boldsymbol{\mathcal{T}}(y)\boldsymbol{V}_t^T(a_t\boldsymbol{I} - \boldsymbol{V}_t\boldsymbol{\mathcal{T}}(y)\boldsymbol{V}_t^T)^{-1}\boldsymbol{\omega}, \qquad \boldsymbol{f}_{in}^t(\boldsymbol{b}) = (\gamma_t\boldsymbol{V}_t^T - \boldsymbol{A}_t)^{-1}\boldsymbol{b}, \tag{70}$$

where $\boldsymbol{\mathcal{T}}(y)$ is the preprocessing function defined in (22) and $a_t$, $\gamma_t$ parameters to be fixed.

We first show that, for suitable choiches of $a_t$ and $\gamma_t$, the non-zero fixed point of this algorithm correspond to an eigenvector of $\boldsymbol{T}$. Dropping the time index for the fixed-point variables and parameters, and defining $\hat{\boldsymbol{G}}_{\boldsymbol{T}} \in \mathbb{R}^{np \times np}$ as

$$\left[ \hat{\boldsymbol{G}}_{\boldsymbol{\mathcal{T}}} \right]_{(i\mu),(j\nu)} \coloneqq \delta_{ij}[\boldsymbol{\mathcal{T}}(\mathrm{y}_i)\boldsymbol{V}^T\left( a_t\boldsymbol{I} - \boldsymbol{V}\boldsymbol{\mathcal{T}}(\mathrm{y}_i)\boldsymbol{V}^T \right)^{-1}]_{\mu\nu}, \quad i, j \in [\![n]\!], \mu\nu \in [\![p]\!], \tag{71}$$

for $i \in [\![n]\!]$, $\mu \in [\![p]\!]$, the fixed points satisfy

$$\Omega_{i\mu} = [\boldsymbol{X}\hat{\mathbf{W}}]_{i\mu} - \sum_{j\in[\![n]\!]} \sum_{\nu\in[\![p]\!]} \underbrace{\sum_{\rho\in[\![p]\!]} [\hat{\boldsymbol{G}}_{\boldsymbol{T}}]_{(i\nu),(j\rho)}V_{\mu\rho}}_{\coloneqq[\hat{\boldsymbol{G}}_{\boldsymbol{T},\boldsymbol{V}}]_{(i\mu),(j\nu)}}\Omega_{j\nu} \tag{72}$$

$$\implies \boldsymbol{\Omega} = \mathrm{mat}\left( \left(\boldsymbol{I}_{np} + \hat{\boldsymbol{G}}_{\boldsymbol{T},\boldsymbol{V}}\right)^{-1} \mathrm{vec}\left(\mathbf{X}\hat{\mathbf{W}}\right) \right), \tag{73}$$

and

$$\hat{\mathbf{W}}(\gamma\boldsymbol{V} - \boldsymbol{A}^T) = \mathbf{X}\mathrm{mat}\left( \hat{\boldsymbol{G}}_{\boldsymbol{T}}\left(\boldsymbol{I}_{np} + \hat{\boldsymbol{G}}_{\boldsymbol{T},\boldsymbol{V}}\right)^{-1} \mathrm{vec}\left(\mathbf{X}\hat{\mathbf{W}}\right) \right) - \hat{\mathbf{W}}\boldsymbol{A}^T \tag{74}$$

$$\implies a\gamma\mathrm{vec}\left(\hat{\mathbf{W}}\boldsymbol{V}\right) = \boldsymbol{T}\mathrm{vec}\left(\hat{\mathbf{W}}\boldsymbol{V}\right), \tag{75}$$

where we used

$$\mathbb{R}^{p\times p} \ni \left[ \hat{\boldsymbol{G}}_{\boldsymbol{T}}\left(\boldsymbol{I}_{np} + \hat{\boldsymbol{G}}_{\boldsymbol{T},\boldsymbol{V}}\right)^{-1} \right]_{(i.),(j.)} = \delta_{ij}\boldsymbol{\mathcal{T}}(\mathrm{y}_i)\boldsymbol{V}^T(a\boldsymbol{I} - \boldsymbol{V}\boldsymbol{\mathcal{T}}(\mathrm{y}_i)\boldsymbol{V}^T)^{-1}$$

$$\left[\left(a\boldsymbol{I} - \boldsymbol{V}\boldsymbol{\mathcal{T}}(\mathrm{y}_i)\boldsymbol{V}^T + \boldsymbol{V}\boldsymbol{\mathcal{T}}(\mathrm{y}_i)\boldsymbol{V}^T\right)\left(a\boldsymbol{I} - \boldsymbol{V}\boldsymbol{\mathcal{T}}(\mathrm{y}_i)\boldsymbol{V}^T\right)^{-1}\right]^{-1}$$

$$= \delta_{ij}a^{-1}\boldsymbol{\mathcal{T}}(\mathrm{y}_i)\boldsymbol{V}^T$$

Therefore, $\hat{\mathbf{W}} = \hat{\mathbf{W}}_{\boldsymbol{T}}\boldsymbol{V}^{-1}$ is a fixed point of the algorithm, for $a_t$ and $\gamma_t$ appropriately chosen, with eigenvalue given by $a_t\gamma_t$ at convergence.[2]

## C.1 State Evolution

The state evolution equations of the overlap matrices are

$$\boldsymbol{M}^{t+1} = \alpha\mathcal{F}(\boldsymbol{M}^t; a_t, \boldsymbol{V}_t) \tag{76}$$

$$\boldsymbol{Q}^{t+1} = \boldsymbol{M}^t(\boldsymbol{M}^t)^T + \alpha(\mathcal{G}(\boldsymbol{M}^t; a_t, \boldsymbol{V}_t) + \tilde{\mathcal{F}}(\boldsymbol{Q}^t; a_t, \boldsymbol{V}_t)) \tag{77}$$

$$\boldsymbol{V}_{t+1} = (\gamma_t\boldsymbol{V}_t^T - \boldsymbol{A}_t)^{-1}, \tag{78}$$

$$\boldsymbol{A}_t = \alpha\mathbb{E}_{\mathrm{y}\sim Z(\mathrm{y})}\left[\boldsymbol{\mathcal{T}}(\mathrm{y})\boldsymbol{V}_t^T(a_t - \boldsymbol{V}_t\boldsymbol{\mathcal{T}}(\mathrm{y})\boldsymbol{V}_t^T)^{-1}\right] \tag{79}$$

with

$$\mathcal{F}(\boldsymbol{M}; a, \boldsymbol{V}) := \mathbb{E}_{\mathrm{y}\sim Z}\left[\boldsymbol{V}\boldsymbol{\mathcal{T}}(\mathrm{y})\boldsymbol{V}^T(a - \boldsymbol{V}\boldsymbol{\mathcal{T}}(\mathrm{y})\boldsymbol{V}^T)^{-1}\boldsymbol{M}\boldsymbol{G}(\mathrm{y})\right],$$

$$\tilde{\mathcal{F}}(\boldsymbol{Q}; a, \boldsymbol{V}) := \mathbb{E}_{\mathrm{y}\sim Z}\left[\boldsymbol{V}\boldsymbol{\mathcal{T}}(\mathrm{y})\boldsymbol{V}^T(a - \boldsymbol{V}\boldsymbol{\mathcal{T}}(\mathrm{y})\boldsymbol{V}^T)^{-1}\boldsymbol{Q}(a - \boldsymbol{V}\boldsymbol{\mathcal{T}}(\mathrm{y})\boldsymbol{V}^T)^{-1}\boldsymbol{V}\boldsymbol{\mathcal{T}}(\mathrm{y})\boldsymbol{V}^T\right]$$

$$\mathcal{G}(\boldsymbol{M}; a, \boldsymbol{V}) := \mathbb{E}_{\mathrm{y}\sim Z}\left[\boldsymbol{V}\boldsymbol{\mathcal{T}}(\mathrm{y})\boldsymbol{V}^T(a - \boldsymbol{V}\boldsymbol{\mathcal{T}}(\mathrm{y})\boldsymbol{V}^T)^{-1}\boldsymbol{M}\boldsymbol{G}(\mathrm{y})\boldsymbol{M}^T(a - \boldsymbol{V}\boldsymbol{\mathcal{T}}(\mathrm{y})\boldsymbol{V}^T)^{-1}\boldsymbol{V}\boldsymbol{\mathcal{T}}(\mathrm{y})\boldsymbol{V}^T\right].$$

Note that $\mathcal{F}(\,\cdot\,; a, \boldsymbol{V})$ is a symmetric linear operator on the space of $p \times p$ matrices, with respect to the inner product $\langle\boldsymbol{M}, \boldsymbol{M}'\rangle := \mathrm{Tr}(\boldsymbol{M}^T\boldsymbol{M}')$:

$$\langle\mathcal{F}(\boldsymbol{M}; a, \boldsymbol{V}), \boldsymbol{M}'\rangle = \mathbb{E}_{\mathrm{y}\sim Z}\mathrm{Tr}\left[\boldsymbol{G}(\mathrm{y})\boldsymbol{M}^T(a - \boldsymbol{V}\boldsymbol{\mathcal{T}}(\mathrm{y})\boldsymbol{V}^T)^{-1}\boldsymbol{V}\boldsymbol{\mathcal{T}}(\mathrm{y})\boldsymbol{V}^T\boldsymbol{M}'\right] \tag{80}$$

$$= \mathbb{E}_{\mathrm{y}\sim Z}\mathrm{Tr}\left[(a - \boldsymbol{V}\boldsymbol{\mathcal{T}}(\mathrm{y})\boldsymbol{V}^T)^{-1}\boldsymbol{V}\boldsymbol{\mathcal{T}}(\mathrm{y})\boldsymbol{V}^T\boldsymbol{M}'\boldsymbol{G}(\mathrm{y})\boldsymbol{M}^T\right] \tag{81}$$

$$= \mathrm{Tr}\mathbb{E}_{\mathrm{y}\sim Z}\left[\boldsymbol{V}\boldsymbol{\mathcal{T}}(\mathrm{y})\boldsymbol{V}^T(a - \boldsymbol{V}\boldsymbol{\mathcal{T}}(\mathrm{y})\boldsymbol{V}^T)^{-1}\boldsymbol{M}'\boldsymbol{G}(\mathrm{y})\boldsymbol{M}^T\right] \tag{82}$$

$$= \langle\mathcal{F}(\boldsymbol{M}'; a, \boldsymbol{V}), \boldsymbol{M}\rangle. \tag{83}$$

This implies that, for $a, \boldsymbol{V}$ fixed, $\mathcal{F}(\,\cdot\,; a, \boldsymbol{V})$ has $p^2$ real eigenvalues $\{\nu_k(a, \boldsymbol{V})\}_{k\in[\![p^2]\!]}$ and admits an orthonormal basis $\{\boldsymbol{M}_k(a, \boldsymbol{V})\}_{k\in[\![p^2]\!]}$ of eigenvectors in $\mathbb{R}^{p\times p}$. Moreover, note that from the state evolution iterations, we can verify that $\boldsymbol{V}_t = \boldsymbol{V}_t^T \implies \boldsymbol{V}_{t+1} = \boldsymbol{V}_{t+1}^T$, therefore, we consider the matrix $\boldsymbol{V}_t$ to be symmetric at all times. From the state evolution equations at convergence

$$\boldsymbol{V} = \sqrt{\gamma^{-1}}\left(\boldsymbol{I} + \alpha\mathbb{E}_{\mathrm{y}\sim Z}\left[\boldsymbol{V}\boldsymbol{\mathcal{T}}(\mathrm{y})\boldsymbol{V}(a - \boldsymbol{V}\boldsymbol{\mathcal{T}}(\mathrm{y})\boldsymbol{V})^{-1}\right]\right)^{1/2} \implies \boldsymbol{V} \succ \boldsymbol{0}. \tag{84}$$

Furthermore, in order to bound the operator norm of $\boldsymbol{V}$, we choose

$$\gamma_t = \|\boldsymbol{V}_t^{-1} + \alpha\mathbb{E}_{\mathrm{y}}\left[\boldsymbol{V}_t\boldsymbol{\mathcal{T}}(\mathrm{y})\boldsymbol{V}_t(a - \boldsymbol{V}_t\boldsymbol{\mathcal{T}}(\mathrm{y})\boldsymbol{V}_t)^{-1}\right]\|_{\mathrm{op}}, \tag{85}$$

so that $\|\boldsymbol{V}_{t\to\infty}\|_{\mathrm{op}} = 1$.
We can distinguish the following cases:

---

[2]Note that the overlap matrices for this algorithm refer to $\hat{\mathbf{W}}_{\boldsymbol{T}}\boldsymbol{V}^{-1}$ and not directly to the spectral estimator itself. However, if $\|\boldsymbol{M}\|_{\mathrm{F}} > 0$, the weak recovery condition is satisfied.

- **Informed fixed points.** (*Theorem 2.8*) Choosing $a_t$ such that

$$\max\left\{ \lim_{t\to\infty} \nu_k(a_t, \boldsymbol{V}_t) : k \in [\![p^2]\!]\right\} = \alpha^{-1}, \tag{86}$$

then $\exists \boldsymbol{M} \neq \boldsymbol{0}$ stable fixed point of the state evolution, and, for $\alpha > \alpha_c$ ($a > 1$)

$$\mathrm{Tr}(\boldsymbol{Q}) = \|\boldsymbol{M}\|_{\mathrm{F}}^2 + \alpha \|\boldsymbol{M}\|_{\mathrm{F}}^2 \mathrm{Tr}\left(\mathcal{G}\left(\frac{\boldsymbol{M}}{\|\boldsymbol{M}\|_{\mathrm{F}}}; a, \boldsymbol{V}\right)\right) + \alpha \mathrm{Tr}(\tilde{\mathcal{F}}(\boldsymbol{Q}; a, \boldsymbol{V})) \implies \tag{87}$$

$$\|\boldsymbol{M}\|_{\mathrm{F}}^2 \geq \mathrm{Tr}(\boldsymbol{Q}) - \alpha \mathrm{Tr}(\tilde{\mathcal{F}}(\boldsymbol{Q}; a, \boldsymbol{V})) \geq \alpha \mathrm{Tr}\left(\mathcal{F}(\boldsymbol{Q}; a, \boldsymbol{V}) - \tilde{\mathcal{F}}(\boldsymbol{Q}; a, \boldsymbol{V})\right) > 0, \tag{88}$$

- **Uninformed fixed points.** (*Theorem 2.9*) Initializing GAMP with $\boldsymbol{M}^0 = \boldsymbol{0}$, which is a fixed point of the state evolution,

$$\boldsymbol{Q} = \alpha \tilde{\mathcal{F}}(\boldsymbol{Q}; a, \boldsymbol{V}),. \tag{89}$$

Similarly to $\mathcal{F}(\ \cdot\ ; a, \boldsymbol{V})$, the symmetric operator $\tilde{\mathcal{F}}(\ \cdot\ ; a, \boldsymbol{V})$ has $p^2$ real eigenvalues. Defining $\nu_1^{\tilde{\mathcal{F}}}(a)$ as its largest one, we notice that $\nu_1^{\tilde{\mathcal{F}}}(1) = \alpha_c^{-1}$ and $\nu_1^{\tilde{\mathcal{F}}}(a \to \infty) \to 0$. Therefore, for $\alpha > \alpha_c$, $\exists a > 1$ such that $\nu_1^{\tilde{\mathcal{F}}} = \alpha^{-1}$ and the state evolution has a fixed point $\boldsymbol{M} = \boldsymbol{0}$, $\boldsymbol{Q} \in \mathbb{S}_+^p \setminus \{\boldsymbol{0}\}$. Moreover, for such $a$, the largest eigenvalue of $\mathcal{F}(\ \cdot\ ; a, \boldsymbol{V})$ is larger than $\alpha^{-1}$, hence the uninformed fixed point is unstable for $\alpha > \alpha_c$. This can be shown repeating a similar argument as the one we have applied in eq. (88). Since the GAMP convergence equations correspond to a generalized power iteration of $\boldsymbol{T}$, normalized by the eigenvalue $a\gamma$, the instability of the uninformed fixed point implies that it is associated to an eigenvector smaller than $\lambda_s$ defined in Theorem 2.8.

## C.2 Sketch of derivation for Conjecture 2.10

These results for the fixed points of the state evolution justify Conjecture 2.10 on the weak recovery properties of the symmetric spectral estimator. The following argument is analogous to the one given for the asymmetric spectral method given in Appendix D. As a consequence of eq. 74, the Algorithm in Definition 2.6 behaves as the power-iteration on the matrix $\boldsymbol{T}$. Suppose that $\lambda_1(\boldsymbol{T}) - \lambda_2(\boldsymbol{T}) = \omega(d^{-\kappa})$ for some $\kappa > 0$. Then, we obtain that $\boldsymbol{w}^t = (a\gamma)^{-1} \boldsymbol{T} \boldsymbol{w}^{t-1}$ converges to the top-most eigenvector of $\frac{1}{a\gamma} \boldsymbol{T}$ in $\mathcal{O}(\log d)$ iterations.

Moreover, based on extensive confirmations in the literature [56, 57, 58], we conjecture that the asymetric spectral method is described by the state-evolution equations up to $\mathcal{O}(\log d)$ iterations.

Theorem 2.8 predicts the convergence of the state evolution iterate $\boldsymbol{M}^t$ to a fixed point corresponding to $\boldsymbol{M} \neq 0$ for $\alpha > \alpha_c$. Since the fixed point conditions for the algorithm stipulate that $\boldsymbol{w}^t = \frac{1}{\lambda_s} \boldsymbol{T} \boldsymbol{w}^t$, we obtain under the validity of the state-evolution description:

$$\frac{1}{d}\left\| \boldsymbol{w}^t - \frac{1}{\lambda_s} \boldsymbol{T} \boldsymbol{w}^t \right\| \xrightarrow[P]{d\to\infty} 0, \tag{90}$$

and consequently

$$\frac{1}{\|\boldsymbol{w}^t\|^2} \boldsymbol{w}^t \boldsymbol{L} \boldsymbol{w}^t \xrightarrow[P]{d\to\infty} \lambda_s. \tag{91}$$

On the other hand, the equivalence to power-iteration implies that the LHS must converge to the top-most eigenvalue of $\boldsymbol{T}$. We thus conclude that:

$$\lambda_1(\boldsymbol{T}) \xrightarrow[P]{d\to\infty} \lambda_s. \tag{92}$$

## C.3 Random Matrix Theory analysis for the case where $\mathbf{G}(y)$ is jointly diagonalizable $\forall y$

*In this section, we will consider the setting of Thm. 2.11 which will analyzed using random matrix theory tools. We first describe in Sec. C.3.1 the setting and how the constants in Conjecture 2.10 simplified in that case and then sketch the outline of the proof using random matrix theory tools in Sec. C.3.2.*

### C.3.1 Introduction to the model

For $l \in [\![p]\!]$ we denote by $z_l = \lambda_l(\mathrm{y})/(\lambda_l(\mathrm{y}) + 1)$ where $\mathrm{y} \sim \mathsf{Z}$ and for $y \in \mathrm{Supp}(\mathsf{Z})$, $\lambda_l(y) \equiv \lambda_l(\boldsymbol{G}(y))$ is the $l$-th eigenvalue of $\boldsymbol{G}(y) = \mathbb{E}_{\mathbf{z} \sim \mathsf{N}(0, \boldsymbol{I}_p)}[\mathbf{z}\mathbf{z}^T - \boldsymbol{I}|\mathrm{y} = y]$. Given the sample $(\mathbf{x}_i, \mathrm{y}_i)_{i \in [\![n]\!]}$, we let $\boldsymbol{D}_l = \mathrm{Diag}(\{\lambda_l(\mathrm{y}_i)(\lambda_l(\mathrm{y}_i) + 1)^{-1}\}_{i \in [\![n]\!]})$ for $l \in [\![p]\!]$. In the rest of this section we will restrict to a particular setting for the model introduced in the main text. We first recall the definition.

**Definition C.1** (Jointly diagonalizable). Let $\boldsymbol{M}(.) : \mathbb{R} \supset I \ni t \mapsto \boldsymbol{M}(t)$ be a symmetric matrix-valued function. We say that $\boldsymbol{M}(.)$ is *jointly diagonalizable* if for all $t \in I$, we have $\boldsymbol{M}(t) = \boldsymbol{U}\boldsymbol{\Lambda}(t)\boldsymbol{U}^T$, with $\boldsymbol{\Lambda}(t)$ diagonal and the orthogonal matrix $\boldsymbol{U}$ is constant with respect to $t$.

We will restrict to a subclass of our model such that one has

**Assumption C.2.** $\boldsymbol{G}(.) : \mathrm{Supp}(\mathsf{Z}) \ni y \mapsto \boldsymbol{G}(y)$ is jointly diagonalizable.

Note that this subset of problems includes many cases of interest, including the ones considered in this manuscript, such as the monomial $g(\boldsymbol{z}) = \prod_{k \in [\![p]\!]} z_k$, the norm $g(\boldsymbol{z}) = p^{-1}\|\boldsymbol{z}\|^2$ and the embedded sparse parity $g(z_1, z_2) = \mathrm{sign}(z_1, z_2)$.

By rotationally invariance of the hidden directions ($\boldsymbol{W}_\star \overset{(d)}{=} \boldsymbol{O}\boldsymbol{W}_\star$ for any orthogonal matrix $\boldsymbol{O}$) two jointly diagonalizable multi-index models specified by the conditional distributions $\mathsf{P}(\cdot|\boldsymbol{z})$ and $\mathsf{P}_{\boldsymbol{O}}(\cdot|\boldsymbol{z}) := \mathsf{P}(\cdot|\boldsymbol{O}\boldsymbol{z})$ are equivalent up to a change basis and in particular share the same $\alpha_c$. Indeed with $\mathsf{Z}_{\boldsymbol{O}}(y) := \mathbb{E}_{\mathbf{z} \sim \mathsf{N}(0, \boldsymbol{I}_p)}\mathsf{P}_{\boldsymbol{O}}(y|\boldsymbol{z})$ one can immediately check that $\mathsf{Z}_{\boldsymbol{O}}(y) = \mathsf{Z}(y)$ and $\mathbb{E}_{\mathbf{z} \sim \mathsf{N}(0, \boldsymbol{I}_p)}[\mathbf{z}\mathsf{P}_{\boldsymbol{O}}(y|\mathbf{z})] = \boldsymbol{O}^T\mathbb{E}_{\mathbf{z} \sim \mathsf{N}(0, \boldsymbol{I}_p)}[\mathbf{z}\mathsf{P}(y|\mathbf{z})]$ and $\mathbb{E}_{\mathbf{z}}[\mathbf{z}\mathbf{z}^T\mathsf{P}_{\boldsymbol{O}}(y|\mathbf{z})] = \boldsymbol{O}^T\mathrm{Cov}[\mathbf{z}|y]\boldsymbol{O}$. As a consequence, we can set $\boldsymbol{G}(y)$ to be diagonal without loss of generality.

Next, we describe how the constants in Conjecture 2.10 simplifies under this jointly diagonalizable setting. To this end, we introduce the convex functions

$$\psi_{\alpha,l}(a) = a\left(1 + \alpha\, \mathbb{E}_{z_l} \frac{z_l}{a - z_l}\right) \quad l \in [\![p]\!], \tag{93}$$

and let $\bar{a}_{\alpha,l} = \arg\min \psi_{\alpha,l}(a)$ be the minima, that is $\bar{a}_{\alpha,l}$ solves

$$\mathbb{E}\left[\left(\frac{z_l}{\bar{a}_{\alpha,l} - z_l}\right)^2\right] = \alpha^{-1} \quad l \in [\![p]\!]. \tag{94}$$

For later use, we also define

$$\zeta_{\alpha,l}(a) = \psi_{\alpha,l}\big(\max(a, \bar{a}_{\alpha,l})\big) \quad l \in [\![p]\!]. \tag{95}$$

we have the following result.

**Lemma C.3.** *Under Assumption C.2, the constants $(\alpha_c, \lambda_b, \lambda_c)$ in Conjecture 2.10 further simplifies to*

(i) $\alpha_c = \min_{l \in [\![p]\!]} 1/(\mathbb{E}\lambda_l(\mathrm{y})^2)$;

(ii) $\lambda_s = 1$,

(iii) $\lambda_b = \max_{l \in [\![p]\!]} \psi_{\alpha,l}(\bar{a}_{\alpha,l})$ .

*Proof.* For Part-(i) since $\boldsymbol{G}(y)$ is diagonal, the critical threshold described in Lem. 1.2 is maximized over rank-one matrices, that is

$$\frac{1}{\alpha_c} = \sup_{\boldsymbol{u} \in \mathbb{S}^{p-1}} \mathbb{E}_{\mathrm{y} \sim \mathsf{Z}}\langle \boldsymbol{u}, \mathrm{Diag}(\{\lambda_l(\mathrm{y})^2\}_{l \in [\![p]\!]})\boldsymbol{u}\rangle, \tag{96}$$

which gives by Courant-Fisher theorem the desired result for the threshold.
We next turn to the analysis of SE of Prop. 2.2 under the jointly diagonalizable assumption C.2. Setting $\mathcal{T}_l(\mathrm{y}) := \lambda_l(\mathrm{y})(\lambda_l(\mathrm{y}) + 1)^{-1}$, the fixed point equations read for any $(\mu, \nu) \in [\![p]\!]^2$:

$$M_{\mu\nu} = \alpha M_{\mu\nu}\mathbb{E}_{\mathrm{y}}\left[\mathcal{T}_\mu(a/V_\mu^2 - \mathcal{T}_\mu(\mathrm{y}))^{-1}\lambda_\nu(\mathrm{y})\right] \tag{97}$$

$$Q_{\mu\mu} = \sum_{\nu\in[\![p]\!]} M_{\mu\nu}^2 + \alpha \left( \mathcal{G}_{\mu\mu}(\boldsymbol{M};a,\boldsymbol{V}) + Q_{\mu\mu}\mathbb{E}_{\mathrm{y}} \left[ \left(\mathcal{T}_\mu(\mathrm{y})(a/V_\mu^2 - \mathcal{T}_\mu(\mathrm{y}))^{-1}\right)^2 \right] \right) \tag{98}$$

$$\gamma V_\mu^2 = 1 + \alpha \mathbb{E}_{\mathrm{y}} \left[ \mathcal{T}_\mu(\mathrm{y})(a/V_\mu^2 - \mathcal{T}_\mu(\mathrm{y}))^{-1} \right] \tag{99}$$

and $\gamma = \max_\mu \left( 1 + \alpha \mathbb{E}_{\mathrm{y}} \left[ \mathcal{T}_\mu(\mathrm{y})(a/V_\mu^2 - \mathcal{T}_\mu(\mathrm{y}))^{-1} \right] \right)$. The candidated for the informative eigenvalues $\lambda_{\mu\nu}^{\boldsymbol{T}} = a_{\mu\nu}\gamma$ of $\boldsymbol{T}$ correspond to the solutions of

$$\alpha^{-1} = \mathbb{E}_{\mathrm{y}} \left[ \mathcal{T}_\mu(\bar{a}_{\mu\nu} - \mathcal{T}_\mu(\mathrm{y}))^{-1}\lambda_\nu(\mathrm{y}) \right], \quad a_{\mu\nu} = \bar{a}_{\mu\nu}V_\mu^2 \tag{100}$$

and are given by

$$\lambda_{\mu\nu}^{\boldsymbol{T}} = a_{\mu\nu}\gamma = \bar{a}_{\mu\nu}\gamma V_\mu^2 = \bar{a}_{\mu\nu} \left( 1 + \alpha \mathbb{E}_{\mathrm{y}} \left[ \mathcal{T}_\mu(\mathrm{y})(\bar{a}_{\mu\nu} - \mathcal{T}_\mu(\mathrm{y}))^{-1} \right] \right) \tag{101}$$

Note that, for $\mu = \nu$, eq. 100 has always a solution for $\alpha > \alpha_{c,\mu} := \mathbb{E}_{\mathrm{y}}[\lambda_\mu(\mathrm{y})^2]$, and $\forall \mu \in [\![p]\!]$

$$\lambda_{\mu\mu}^{\boldsymbol{T}} = \bar{a}_{\mu\mu} \left( 1 + \alpha \mathbb{E}_{\mathrm{y}} \left[ \mathcal{T}_\mu(\mathrm{y})(\bar{a}_{\mu\mu} - \mathcal{T}_\mu(\mathrm{y}))^{-1} \right] \right) \tag{102}$$

$$= \bar{a}_{\mu\mu} \left( 1 + \alpha \mathbb{E}_{\mathrm{y}} \left[ \mathcal{T}_\mu(\mathrm{y})(\bar{a}_{\mu\mu} - \mathcal{T}_\mu(\mathrm{y}))^{-1} \right] - \frac{1}{\bar{a}_{\mu\mu}} \underbrace{\mathbb{E}_{\mathrm{y}}[\lambda_\mu(\mathrm{y})]}_{=0} \right) \tag{103}$$

$$= \bar{a}_{\mu\mu} \left( 1 - \alpha \frac{\bar{a}_{\mu\mu} - 1}{\bar{a}_{\mu\mu}} \underbrace{\mathbb{E}_{\mathrm{y}} \left[ \mathcal{T}_\mu(\mathrm{y})\lambda_\mu(\mathrm{y})(\bar{a}_{\mu\mu} - \mathcal{T}_\mu(\mathrm{y}))^{-1} \right]}_{1/\alpha} \right) \tag{104}$$

$$= 1 \tag{105}$$

These eigenvalues are informative as

$$\sum_\nu M_{\mu\nu}^2 > Q_{\mu\mu} \left( 1 - \alpha \mathbb{E}_{\mathrm{y}} \left[ \left(\mathcal{T}_\mu(\mathrm{y})(\bar{a}_{\mu\mu} - \mathcal{T}_\mu(\mathrm{y}))^{-1}\right)^2 \right] \right) \tag{106}$$

$$> Q_{\mu\mu} \left( 1 - \alpha \underbrace{\mathbb{E}_{\mathrm{y}} \left[ \mathcal{T}_\mu(\mathrm{y})\lambda_\mu(\mathrm{y})(\bar{a}_{\mu\mu} - \mathcal{T}_\mu(\mathrm{y}))^{-1} \right]}_{=\alpha^{-1}} \right) = 0 \tag{107}$$

Analogously, the edge of the bulk for each block of $\boldsymbol{T}$ can be obtained solving, for $\alpha > \alpha_{c,\mu}$

$$\alpha^{-1} = \mathbb{E}_{\mathrm{y}} \left[ \left( \frac{\mathcal{T}_\mu(\mathrm{y})}{\hat{a}_\mu - \mathcal{T}_\mu(\mathrm{y})} \right)^2 \right], a_\mu^b = \hat{a}_\mu V_\mu^2, \tag{108}$$

and the correspondent eigenvalues $\lambda_\mu^{\boldsymbol{T},b}$ are given by

$$\lambda_\mu^{\boldsymbol{T},b} = a_\mu^b \gamma = \hat{a}_\mu \left( 1 + \alpha \mathbb{E}_{\mathrm{y}} \left[ \frac{\mathcal{T}_\mu(\mathrm{y})}{\hat{a}_\mu - \mathcal{T}_\mu(\mathrm{y})} \right] \right) \tag{109}$$

$$= \hat{a}_\mu \left( 1 - \alpha \frac{\hat{a}_\mu - 1}{\hat{a}_\mu} \mathbb{E}_{\mathrm{y}} \left[ \frac{\mathcal{T}_\mu(\mathrm{y})\lambda_\mu(\mathrm{y})}{\hat{a}_\mu - \mathcal{T}_\mu(\mathrm{y})} \right] \right) \tag{110}$$

$$< \hat{a}_\mu \left( 1 - \alpha \frac{\hat{a}_\mu - 1}{\hat{a}_\mu} \underbrace{\mathbb{E}_{\mathrm{y}} \left[ \left( \frac{\lambda_\mu(\mathrm{y})}{(\hat{a}_\mu - 1)\lambda_\mu(\mathrm{y}) + \hat{a}_\mu} \right)^2 \right]}_{\alpha^{-1}} \right) \tag{111}$$

$$= 1 = \lambda_{\mu\mu}^{\boldsymbol{T}} \tag{112}$$

from which we conclude Part-(ii) and (iii) of the Lemma. $\square$

### C.3.2 Outline of the proof using RMT

We give a proof of Conjecture 2.10 with the values for the threshold $\alpha_c$, the top outlier $\lambda_s = 1$ and the edge $\lambda_b$ computed in Lem. C.3.

Assuming C.2, the symmetric spectral estimator introduced in Eq. (13) has the block-diagonal structure

$$
\boldsymbol{T} = \begin{pmatrix} \mathbf{X}^T \boldsymbol{D}_1 \mathbf{X} & \mathbf{0}_{d \times d} & \dots & \mathbf{0}_{d \times d} \\ \mathbf{0}_{d \times d} & \mathbf{X}^T \boldsymbol{D}_2 \mathbf{X} & \ddots & \vdots \\ \vdots & \ddots & \ddots & \mathbf{0}_{d \times d} \\ \mathbf{0}_{d \times d} & \dots & \mathbf{0}_{d \times d} & \mathbf{X}^T \boldsymbol{D}_p \mathbf{X} \end{pmatrix}, \tag{113}
$$

and thus its spectral properties can be immediately obtained from its diagonal block since we have

$$
\operatorname{Spec}(\boldsymbol{T}) = \bigcup_{l \in [\![p]\!]} \operatorname{Spec}(\tilde{\boldsymbol{T}}_l) \quad \text{where} \quad \tilde{\boldsymbol{T}}_l := \mathbf{X}^T \boldsymbol{D}_l \mathbf{X}. \tag{114}
$$

In particular, the top eigenvalue of $\boldsymbol{T}$ appearing in Conjecture 2.10 is simply the max of the top eigenvalue of each block $\tilde{\boldsymbol{T}}_l$ and one can restrict to the study of the spectral properties of the $\tilde{\boldsymbol{T}}_l$, following closely the derivation of [55] which tackles the case $p = 1$.

For each $\tilde{\boldsymbol{T}}_l$, we will partition the column of the sensing vector $\boldsymbol{x}_i$ into parts that align with the hidden components $\boldsymbol{w}_\star$ and part that lives in the orthogonal complement, the only difference with the setting considered in [55] being in that now one has to deal with $p$ hidden directions instead of one. To do so, we first use the Gram-Schmidt decomposition of the hidden matrix $\frac{1}{\sqrt{d}} \boldsymbol{W}_\star$:

$$
\frac{1}{\sqrt{d}} \boldsymbol{W}_\star = \boldsymbol{V} \boldsymbol{R} \tag{115}
$$

where $\boldsymbol{V} = (\boldsymbol{v}_1, \dots, \boldsymbol{v}_p)$ is a semi-orthogonal matrix of dimension $(d \times p)$ and $R_{ij} = \langle \boldsymbol{u}_i, \boldsymbol{w}_j \rangle \delta_{i \leq j} \in \mathbb{R}^{p \times p}$ is upper triangular. Note that as $\boldsymbol{w}_{i,\star}$ are iid standard Gaussian, as $d \to \infty$ we have $R_{ii} \to 1$ and $R_{ij} \to 0$ for $i \neq j$, exponentially fast. By rotationally invariance one has $\mathbf{X} \stackrel{(d)}{=} \boldsymbol{O} \mathbf{X}$ for any $\boldsymbol{O}$ orthogonal matrix, hence one can fix $\boldsymbol{v}_i = \boldsymbol{e}_i$ (the $i$-th canonical vector) without loss of generality. We decompose each vectors $\boldsymbol{x}_i$ in this basis as follows

$$
\mathbf{x}_i = (\boldsymbol{\kappa}_i \quad \mathbf{u}_i)^T \text{ with } \boldsymbol{\kappa}_i \in \mathbb{R}^p \text{ and } \boldsymbol{u}_i \in \mathbb{R}^{d-p} \tag{116}
$$

or equivalently

$$
\mathbf{X} = \begin{pmatrix} \boldsymbol{K} \\ \mathbf{U} \end{pmatrix}^T \text{ with } \boldsymbol{K} := \begin{pmatrix} \boldsymbol{\kappa}_1 \\ \vdots \\ \boldsymbol{\kappa}_p \end{pmatrix} \in \mathbb{R}^{p \times n} \text{ and } \mathbf{U} := \begin{pmatrix} \mathbf{u}_1 \\ \vdots \\ \mathbf{u}_p \end{pmatrix} \in \mathbb{R}^{(d-p) \times n} \tag{117}
$$

where $\boldsymbol{K}$ is by construction a matrix with iid Gaussian entries. From this partition, we can re-write each $\tilde{\boldsymbol{T}}_l$ as

$$
\tilde{\boldsymbol{T}}_l = \begin{pmatrix} \boldsymbol{R}_l & \boldsymbol{Q}_l^T \\ \boldsymbol{Q}_l & \mathbf{P}_l \end{pmatrix} \in \mathbb{R}^{d \times d} \quad \text{with} \quad \begin{cases} \boldsymbol{R}_l & := \frac{1}{d} \boldsymbol{K}^T \boldsymbol{D}_l \boldsymbol{K} \in \mathbb{R}^{p \times p}, \\ \boldsymbol{Q}_l & := \frac{1}{d} \mathbf{U}(\boldsymbol{D}_l \boldsymbol{K}^\top) \in \mathbb{R}^{(d-p) \times p}, \\ \mathbf{P}_l & := \frac{1}{d} \mathbf{U}^T \boldsymbol{D}_l \mathbf{U} \in \mathbb{R}^{(d-p) \times (d-p)}. \end{cases} \tag{118}
$$

**Lemma C.4** (Edge and existence of outliers)**.** *As $d \to \infty$, the empirical spectral distribution of $\boldsymbol{T}_l$ converges weakly almost surely to a distribution $\mu_l$ with rightmost edge $\tau_l := \psi_{\alpha,l}(\bar{a}_{\alpha,l})$. Furthermore, for each $l \in [\![p]\!]$, there exists up to $p$ outliers in the spectrum of $\boldsymbol{T}_l$ above this edge $\tau_l$.*

*Proof.* This follows from the same proof of Proposition 3.1 in [55] (see Appendix A.4): by eigenvalue interlacing theorem, if put the eigenvalues in increasing order, we have for any $i \in [\![(d-p)p]\!]$: $\lambda_i^{(\nearrow)}(\tilde{\boldsymbol{T}}_l) \leq \lambda_i^{(\nearrow)}(\boldsymbol{P}_l) \leq \lambda_{i+p}^{(\nearrow)}(\tilde{\boldsymbol{T}}_l)$. Furthermore the empirical distribution of $\boldsymbol{P}_l$ converges *strongly* to a limiting distribution $\mu_l$ with rightmost edge $\tau_l$, since all but the top $p$ eigenvalues of $\boldsymbol{T}_l$ are trapped between eigenvalues of $\boldsymbol{P}_l$, one gets the desired result. $\square$

From Eq. (114), one immediately obtains that the empirical distribution of the full matrix $\boldsymbol{T}$ converges weakly almost surely to the distribution $\mu := \frac{1}{p} \sum_l \mu_l$ whose rightmost edge is given by $\lambda_b$ in Lem. C.3. Next, following again [55], we map the position of the top outliers in each block $\boldsymbol{T}_l$ to an additive spiked matrix model.

**Lemma C.5.** *Let $\tilde{\boldsymbol{P}}_l(\mu) = \boldsymbol{P}_l + \boldsymbol{Q}_l(\boldsymbol{R}_l - \mu\boldsymbol{I})^{-1}\boldsymbol{Q}_l^T$ denotes the family of spiked matrices indexed by a parameter $\mu \in \mathbb{R} \setminus \mathrm{Spec}(\boldsymbol{R}_l)$ and set the function $L_l(\mu) := \lambda_1(\tilde{\boldsymbol{P}}_l(\mu))$, where $\lambda_1$ denotes the highest eigenvalue, then we have $\lambda_1(\tilde{\boldsymbol{T}}_l) = L_l(\mu^\star)$, where $\mu^\star$ is the unique fixed point of the equation $L_l(\mu) = \mu$.*

*Proof.* This follows from the determinant lemma in the same proof of Proposition 3.1 of [55] (see Section 3.2) with their $(1 \times 1)$ upper block replaced now by the $(p \times p)$ block $\boldsymbol{R}_l$. $\qquad\square$

**Lemma C.6.** *Let $\tilde{\boldsymbol{P}}_l(\mu)$ as in Lem.C.5, then as $d \to \infty$ if the largest real solution (in $a$) of*

$$\det\left(\mu\boldsymbol{I} - \mathbb{E}_{(\boldsymbol{\kappa},\mathsf{z}_l)}\left\{\frac{a\mathsf{z}_l}{a - \mathsf{z}_l}\boldsymbol{\kappa}\boldsymbol{\kappa}^T\right\}\right) = 0 \qquad l \in [\![p]\!]. \tag{119}$$

*exists and we denote it by $a_{1,l}$, we have $L_l(\mu) \to a_{1,l}$ and otherwise $L_l(\mu) \to \tau_l$.*

*Proof.* Applying the determinant lemma to the matrix $\tilde{\boldsymbol{P}}_l(\mu)$, one finds that $L_l(\mu)$ is characterized as the largest solution of the equation

$$\det\left(\mu - \boldsymbol{R}_k - \boldsymbol{Q}_k^T(L_l(\mu)\boldsymbol{I} - \mathbf{P}_k)^{-1}\boldsymbol{Q}_k\right) = 0. \tag{120}$$

Moreover, by the law of large numbers, we have the following almost sure limits as $d \to \infty$:

$$\boldsymbol{R}_k \xrightarrow[d\to\infty]{\text{a.s.}} \mathbb{E}_{(\boldsymbol{\kappa},\mathsf{z}_l)}\mathsf{z}_l\boldsymbol{\kappa}\boldsymbol{\kappa}^T\,, \tag{121}$$

$$\boldsymbol{Q}_k^T(\lambda\boldsymbol{I}_{d-p} - \mathbf{P}_k)^{-1}\boldsymbol{Q}_k \xrightarrow[d\to\infty]{\text{a.s.}} \mathbb{E}_{(\boldsymbol{\kappa},\mathsf{z}_l)}\frac{\mathsf{z}_l^2}{\lambda - \mathsf{z}_l}\boldsymbol{\kappa}\boldsymbol{\kappa}^T\,, \tag{122}$$

where $\boldsymbol{\kappa} \sim \mathsf{N}(\boldsymbol{0}, \boldsymbol{I}_p)$. Substituting these limits into the determinant equation yields the asymptotic equation for the position of the top outlier $L(\mu)$. $\qquad\square$

Combing Lemma C.5 with Lemma C.6, we can characterize the limiting position of the top outlier of $\boldsymbol{T}_l$ in terms of $\zeta_{\alpha,l}$: If the largest real solution (in $a$) of

$$\det\left(\zeta_{\alpha,l}(a)\boldsymbol{I} - \alpha\mathbb{E}_{(\boldsymbol{\kappa},\mathsf{z}_l)}\left\{\frac{a\mathsf{z}_l}{a - \mathsf{z}_l}\boldsymbol{\kappa}\boldsymbol{\kappa}^T\right\}\right) = 0 \qquad l \in [\![p]\!]\,, \tag{123}$$

exists and call it by $a_{\alpha,l}^*$ then we have

$$\lambda_1(\boldsymbol{T}_l) \to \zeta_{\alpha,l}(a_{\alpha,l}^*)\,. \tag{124}$$

Otherwise, if there is no such solution to Eq.(123), we have no outliers, that is

$$\lambda_1(\boldsymbol{T}_l) \to \tau_l\,. \tag{125}$$

As a consequence, the critical threshold $\alpha_c$ for the appearance of an outlier in the full matrix $\boldsymbol{T}$ is given as the minimal value of $\alpha$ for which there exists a solution (in $a$) of Eq. (123) as $l \in [\![p]\!]$. To obtain explicitly this threshold, we first express Eq. (123) in terms of the $\boldsymbol{G}(y) = \mathbb{E}_{\mathbf{z}}\{\boldsymbol{\kappa}\boldsymbol{\kappa}^T - \boldsymbol{I}|\mathsf{y} = y\}$:

$$\det\left(\zeta_{\alpha,l}(a)\boldsymbol{I} - a\alpha\mathbb{E}_{\mathsf{y}}\frac{\lambda_l(\mathsf{y})(\lambda_l(\mathsf{y}) + 1)^{-1}}{a - \lambda_l(\mathsf{y})(\lambda_l(\mathsf{y}) + 1)^{-1}}\boldsymbol{G}(\mathsf{y}) - a\alpha\mathbb{E}_{\mathsf{z}_l}\frac{\mathsf{z}_l}{a - \mathsf{z}_l}\right) = 0 \qquad l \in [\![p]\!], \tag{126}$$

where we have replaced $\mathsf{z}_l$ by its expression in the first expectation. Assuming now that $\alpha$ is such that there exists a solution $a_{\alpha,l}^* > \bar{a}_{\alpha,l}$ of Eq. (126), by definition (95) of $\zeta_{\alpha,l}(.)$, the latter can be replaced by $\psi_{\alpha,l}$. Next from Assumption C.2, $\boldsymbol{G}(y)$ is diagonal with eigenvalues $\{\lambda_l(y)\}_{l\in[\![p]\!]}$ and thus, each solution of Eq. (126) must solve for $l' \in [\![p]\!]$:

$$\mathbb{E}\left[\frac{\lambda_l(\mathsf{y})\lambda_{l'}(\mathsf{y})(\lambda_l(\mathsf{y}) + 1)^{-1}}{a - \lambda_l(\mathsf{y})(\lambda_l(\mathsf{y}) + 1)^{-1}}\right] = \frac{1}{\alpha}; \tag{127}$$

and the largest one is given for $l' = l$ that is

$$\mathbb{E}\left[\frac{\lambda_l(\mathsf{y})^2(\lambda_l(\mathsf{y}) + 1)^{-1}}{a - \lambda_l(\mathsf{y})(\lambda_l(\mathsf{y}) + 1)^{-1}}\right] = \frac{1}{\alpha}; \tag{128}$$

and always exists for as long as $\alpha > \alpha_{c,l} := \left(\mathbb{E}[\lambda_l(\mathrm{y})^2]\right)^{-1}$, from which one gets the desired result since $\alpha_c = \min_{l \in [\![p]\!]} \alpha_{c,l}$.

To conclude, we need to show that for $\alpha > \alpha_c$, the asymptotic of $\max_{l \in [\![p]\!]} \lambda_1(\boldsymbol{T}_l)$ given by Eq. (124), further simplifies to $\max_{l \in [\![p]\!]} \lambda_1(\boldsymbol{T}_l) \to 1$. Replacing $\zeta_{\alpha,l}$ by its expression, we have :

$$\zeta_{\alpha,l}(a) = a \left( 1 + \alpha\, \mathbb{E}_{\mathrm{y}} \frac{\lambda_l(\mathrm{y})(\lambda_l(\mathrm{y}) + 1)^{-1}}{a - \lambda_l(\mathrm{y})(\lambda_l(\mathrm{y}) + 1)^{-1}} \right) \tag{129}$$

since by assumption of our model, we must have $\mathbb{E}_{\mathrm{y}} \lambda_l(\mathrm{y}) = 0$, the latter can also be expressed as

$$\zeta_{\alpha,l}(a) = a \left( 1 - \alpha \frac{a-1}{a} \mathbb{E}_{\mathrm{y}} \left[ \frac{\lambda_l(\mathrm{y})^2(\lambda_l(\mathrm{y}) + 1)^{-1}}{a - \lambda_l(\mathrm{y})(\lambda_l(\mathrm{y}) + 1)^{-1}} \right] \right) , \tag{130}$$

and since for $\alpha > \alpha_c$, the largest solution $a^\star$ must solve Eq.(128), the expectation reduces to $1/\alpha$ such that one has

$$\max_{l \in [\![p]\!]} \zeta_{\alpha,l}(a^*_{\alpha,l}) = 1 \, , \tag{131}$$

which concludes the proof.

# D   Sketch of the derivation of Conjecture 2.5

As we saw in Equations 55, the Algorithm in Definition 2.1 is equivalent to power-iteration on the matrix $\boldsymbol{L}$. We may further suppose that $\mathrm{Re}(\lambda_1(\boldsymbol{L})) - \mathrm{Re}(\lambda_2(\boldsymbol{L})) = \omega(d^{-\kappa})$ for some $\kappa > 0$. For instance, $\kappa = \frac{2}{3}$ under the Tracy-Widom scalings.

Assuming such a scaling for the spectral gap, we obtain that $\boldsymbol{\omega}^t = \gamma^{-1} \boldsymbol{L} \boldsymbol{\omega}^{t-1}$ converges to the top-most eigenvector of $\frac{1}{\gamma} \boldsymbol{L}$ in $\mathcal{O}(\log d)$ iterations.

Moreover, based on extensive confirmations in the literature [56, 57, 58], we conjecture that the asymetric spectral method is described by the state-evolution equations up to $\mathcal{O}(\log d)$ iterations.

Theorems 2.3 and 2.4 predict the convergence of the state evolution iterate $\boldsymbol{M}^t$ to a fixed point corresponding to $\boldsymbol{M} = 0$ for $\alpha < \alpha_c$ and $\boldsymbol{M} \neq 0$ for $\alpha > \alpha_c$. Since the fixed point conditions for the algorithm stipulate that $\omega^t = \frac{1}{\gamma_s} \boldsymbol{L} \omega^t$, we obtain under the validity of the state-evolution description:

$$\frac{1}{d} \left\| \boldsymbol{\omega}^t - \frac{1}{\gamma_s} \boldsymbol{L} \boldsymbol{\omega}^t \right\| \xrightarrow[P]{d \to \infty} 0, \tag{132}$$

and consequently

$$\frac{1}{\|\boldsymbol{\omega}^t\|^2} \boldsymbol{\omega}^t \boldsymbol{L} \boldsymbol{\omega}^t \xrightarrow[P]{d \to \infty} \gamma_s. \tag{133}$$

On the other hand, the equivalence to power-iteration implies that the LHS must converge to the top-most eigenvalue of $\boldsymbol{L}$. We thus conclude that:

$$\lambda_1(\boldsymbol{L}) \xrightarrow[P]{d \to \infty} \gamma_s. \tag{134}$$

# E   Details on examples - Asymmetric spectral method

## E.1   Single-index models

The case of single-index models ($p = 1$) allows for significant simplifications, as

$$\mathcal{F}(M \in \mathbb{R}) = M \mathbb{E}_{\mathrm{y} \sim \mathsf{Z}} \left[ \left( \mathrm{Var}[z|y] - 1 \right)^2 \right], \tag{135}$$

$$\mathcal{G}(M \in \mathbb{R}) = M^2 \mathbb{E}_{\mathrm{y} \sim \mathsf{Z}} \left[ \left( \mathrm{Var}[z|y] - 1 \right)^3 \right]. \tag{136}$$

This leads to the well known expression for the critical weak recovery threshold [12, 9, 55, 10, 29]

$$\alpha_c^{-1} = \mathbb{E}_{y \sim \mathsf{Z}}\left[\left(\mathrm{Var}[z|y] - 1\right)^2\right], \tag{137}$$

and to the following result for the overlap parameter $m = M/\sqrt{Q}$

$$m^2 = \begin{cases} (\alpha - \alpha_c)\left(\alpha + \alpha_c^2 \mathbb{E}_{y \sim \mathsf{Z}}\left[\left(\mathrm{Var}[z|y] - 1\right)^3\right]\right)^{-1}, & \alpha \geq \alpha_c \\ 0, & \alpha < \alpha_c \end{cases}. \tag{138}$$

### E.2 $\boldsymbol{G}(y)$ jointly diagonalizable $\forall y$

Following the arguments at the beginning of Appendix C.3, whenever we are in this setting, we can consider without loss of generality $\boldsymbol{G}(y) = \mathbb{E}[\mathbf{z}\mathbf{z}^T - \boldsymbol{I}|\mathsf{y} = y] = \mathrm{diag}(\lambda_1(y), \ldots, \lambda_p(y))$. The eigenpairs of the operator $\mathcal{F}$ eq. (63) are given by

$$\nu_{(k,h)} = \mathbb{E}_y[\lambda_k(y)\lambda_h(y)], \quad \boldsymbol{M}_{(k,h)} \text{ with } [\boldsymbol{M}_{(k,h)}]_{\mu\nu} = \delta_{k\mu}\delta_{h\nu}, \quad \forall k, h \in [\![p]\!], \tag{139}$$

with the critical sample complexity $\alpha_c^{-1} = \nu_1 := \max_{k \in [\![p]\!]} \nu_{(k,k)}$ [3]. If the maximum is achieved by more than one pair of indices, we expect that the matrix $\boldsymbol{L}$ principal eigenvalue is degenerate, with degeneracy given by the cardinality of the set $\mathcal{I} = \{(k, h) \in \{1, ..., p\}^2 | \nu_{(k,h)} = \alpha_c^{-1}\}$. Note that $\forall k, h, \nu_{(k,h)} = \nu_{(h,k)}$ and if $\nu_{(k,h)} = \max_{\mu \in \{k,h\}} \nu_{\mu\mu} \implies \nu_{(k,k)} = \nu_{(h,h)}$.[4]
The generic principal eigenvector of $\mathcal{F}$ is given by $\boldsymbol{M} = \|\boldsymbol{M}\|_F \sum_{(k,h) \in \mathcal{I}} c_{(k,h)} \boldsymbol{M}_{(k,h)}$ with $\sum c_{(k,h)}^2 = 1$. We introduce the ansatz: $\boldsymbol{Q}$ s.t. $\boldsymbol{Q}_{kk} \neq 0$ iff $(k, k) \in \mathcal{I}$; this implies $\mathrm{Tr}(\mathcal{F}(\boldsymbol{Q})) = \sum_k \mathbb{E}[\lambda_k(y)^2] Q_{kk} = \alpha_c^{-1}\mathrm{Tr}(\boldsymbol{Q})$. Eq. (66) becomes

$$\mathrm{Tr}(\boldsymbol{Q}) = \|\boldsymbol{M}\|_F^2 \left(1 + \frac{\alpha_c^2}{\alpha} \sum_{(k,h) \in \mathcal{I}} c_{(k,h)}^2 \mathrm{Tr}(\mathcal{G}(\boldsymbol{M}_{(k,h)}))\right) + \frac{\alpha_c^2}{\alpha}\mathrm{Tr}(\mathcal{F}(\boldsymbol{Q})) \implies \tag{140}$$

$$\mathrm{Tr}(\boldsymbol{Q}) = \|\boldsymbol{M}\|_F^2 \left(1 + \frac{\alpha_c^2}{\alpha} \sum_{(k,h) \in \mathcal{I}} c_{(k,h)}^2 \mathbb{E}_y[\lambda_k(y)^2 \lambda_h(y)]\right) + \frac{\alpha_c}{\alpha}\mathrm{Tr}(\boldsymbol{Q}) \implies \tag{141}$$

$$\frac{\|\boldsymbol{M}\|_F^2}{\mathrm{Tr}(\boldsymbol{Q})} = \left(1 - \frac{\alpha_c}{\alpha}\right)\left(1 + \frac{\alpha_c^2}{\alpha} \sum_{(k,h) \in \mathcal{I}} c_{(k,h)}^2 \mathbb{E}_y[\lambda_k(y)^2 \lambda_h(y)]\right)^{-1} \tag{142}$$

As a special case, we consider the example $\lambda_k(y) = \lambda_h(y)$ for all $h, k$ such that $\nu_{(k,k)} = \nu_{(h,h)} = \nu_1$. One instance for this case is given by the link function $g(\boldsymbol{z}) = p^{-1} \sum_{k \in [\![p]\!]} z_k^2$. Then, defining $\lambda^{(3)} = \mathbb{E}_{\mathsf{y}}[\lambda_k(\mathsf{y})^3]$ for any $k|(k, k) \in \mathcal{I}$, the solution for $m^2$ does not depend on the coefficients $c_{(k,h)}$ and simplifies to

$$\frac{\|\boldsymbol{M}\|_F^2}{\mathrm{Tr}(\boldsymbol{Q})} = \left(1 - \frac{\alpha_c}{\alpha}\right)\left(1 + \frac{\alpha_c^2}{\alpha}\lambda^{(3)}\right)^{-1}. \tag{143}$$

The above expression does not depend on the coefficients $c_{(k,h)}$, therefore it is valid for all the degenerate directions in the principal eigenspace.
We consider now specific cases of link functions that are such that $\mathrm{Cov}[\mathbf{z}|y]$ is jointly diagonalizable $\forall y$. We refer to [18] for the derivation of the expressions of $\mathsf{Z}(y)$ and $\mathrm{Cov}[\mathbf{z}|y]$ in all the examples contained in this Appendix E.2.
Note that, in general, if $\mathsf{P}(\cdot|\boldsymbol{z}) = \mathsf{P}(\cdot|\boldsymbol{Qz})$, for $\boldsymbol{Q}$ orthogonal, then $\boldsymbol{G}(\cdot) = \boldsymbol{Q}\boldsymbol{G}(\cdot)\boldsymbol{Q}^T$. We use this property to find examples of jointly diagonalizable $\boldsymbol{G}(\cdot)$. Given any $\mathsf{P}(\cdot|\boldsymbol{z})$ depending on $\boldsymbol{z}$ through a quadratic form $\boldsymbol{z}^T \boldsymbol{A} \boldsymbol{z} = 1/2\, \boldsymbol{z}^T(\boldsymbol{A} + \boldsymbol{A}^T)\boldsymbol{z}$, with $\boldsymbol{A}$ deterministic, and $\boldsymbol{U}$, whose columns $\boldsymbol{u}_\mu$ are

---

[3]Note that $\forall k, h \in [\![p^2]\!]$, $\nu_{(k,h)} \leq \max(\nu_{(k,k)}, \nu_{(h,h)})$.

[4]Without loss of generality $\nu_{(h,h)} \leq \nu_{(k,k)}$ and $\nu_{(k,k)} = \nu_{(k,k)} \leq \sqrt{\nu_{(k,k)}\nu_{(h,h)}} \implies \nu_{(k,k)} \leq \nu_{(h,h)}$. Therefore $\nu_{(k,k)} = \nu_{(h,h)}$.

orthonormal eigenvectors of $\frac{1}{2}\left(\boldsymbol{A}+\boldsymbol{A}^T\right)$, such models are invariant with respect to the orthogonal matrix $\boldsymbol{Q}_\nu = \boldsymbol{U}\operatorname{diag}\left(\left((-1)^{2-\delta_{\mu,\nu}}\right)_{\mu\in[\![p]\!]}\right)\boldsymbol{U}^T$, for any $\nu \in [\![p]\!]$. Therefore,

$$\boldsymbol{G}(y) = \boldsymbol{Q}_\nu \boldsymbol{G}(y)\boldsymbol{Q}_\nu^T \tag{144}$$

$$\implies \boldsymbol{u}_\mu \boldsymbol{G}(y)\boldsymbol{u}_\nu = \boldsymbol{u}_\mu \boldsymbol{Q}_\nu \boldsymbol{G}(y)\boldsymbol{Q}_\nu^T \boldsymbol{u}_\nu \tag{145}$$

$$\implies \boldsymbol{u}_\mu \boldsymbol{G}(y)\boldsymbol{u}_\nu = -\boldsymbol{u}_\mu \boldsymbol{G}(y)\boldsymbol{u}_\nu, \quad \mu \neq \nu \tag{146}$$

$$\implies \boldsymbol{u}_\mu \boldsymbol{G}(y)\boldsymbol{u}_\nu = 0, \quad \mu \neq \nu. \tag{147}$$

Hence, $\boldsymbol{G}(y)$ is jointly diagonalizable with respect to the basis of eigenvectors $\boldsymbol{U}$. Similarly, we can show that any model $\mathsf{P}(\cdot|\boldsymbol{z})$ depending on $\boldsymbol{z}$ through $\prod_{\mu\in[\![p]\!]} z_\mu$ correspond to a jointly diagonalizable $\boldsymbol{G}(\cdot)$. In fact, if $p = 2$, it depends on a quadratic form, while if $p > 3$ it is invariant with respect to $\boldsymbol{Q}_{\nu\kappa} = \operatorname{diag}\left(\left((-1)^{2-\delta_{\mu,\nu}-\delta_{\mu,\kappa}}\right)_{\mu\in[\![p]\!]}\right)$, for any $\nu \neq \kappa$. As in the previous example, we can show that $G_{\mu\nu}(y) = 0$, for any $\mu \neq \nu$ and $\boldsymbol{G}(y)$ is diagonal.

**E.2.1** $g(z_1,\ldots,z_p) = p^{-1}\sum_{k\in[\![p]\!]} z_k^2$

$$\mathsf{Z}(y) = \frac{e^{-\frac{py}{2}}}{y2^{p/2}\Gamma\left(\frac{p}{2}\right)}(py)^{p/2}, \qquad \mathbb{E}[\mathbf{zz}^T|\mathrm{y}] = y\boldsymbol{I}. \tag{148}$$

For a generic $\boldsymbol{M} \in \mathbb{R}^{p\times p}$

$$\mathcal{F}(\boldsymbol{M}) = \boldsymbol{M}\int_0^\infty \mathsf{Z}(y)(y-1)^2\mathrm{d}y = \frac{2}{p}\boldsymbol{M}, \tag{149}$$

$$\mathcal{G}(\boldsymbol{M}) = \boldsymbol{M}\boldsymbol{M}^T\int_0^\infty \mathsf{Z}(y)(y-1)^3\mathrm{d}y = \frac{8}{p^2}\boldsymbol{M}\boldsymbol{M}^T, \tag{150}$$

therefore $\alpha_c = p/2$ and $\lambda^{(3)} = 8/p^3$. Plugging these quantities in eq. (143), the overlap matrices at convergence satisfy

$$\frac{\|\boldsymbol{M}\|_{\mathrm{F}}^2}{\operatorname{Tr}(\boldsymbol{Q})} = \begin{cases} \frac{1}{2}(2\alpha - p)(\alpha+2)^{-1} & \alpha \geq p/2 \\ 0 & \alpha < p/2 \end{cases} \tag{151}$$

**E.2.2** $g(z_1, z_2) = \operatorname{sign}(z_1 z_2)$

$$\mathsf{Z}(y) = \frac{1}{2}, \qquad \mathbb{E}[\mathbf{zz}^T|\mathrm{y}] = \frac{2y}{\pi}\begin{pmatrix} 0 & 1 \\ 1 & 0 \end{pmatrix} + \boldsymbol{I}. \tag{152}$$

The matrix $\mathbb{E}[\mathbf{zz}^T - \boldsymbol{I}|\mathrm{y}]$ is jointly diagonalizable $\forall y$, with eigenvalues $\lambda_1(y) = 2\mathrm{y}\pi^{-1}$ and $\lambda_2(y) = -2\mathrm{y}\pi^{-1}$. Therefore, the eigenvalues of $\mathcal{F}$ are given by

$$\alpha_c^{-1} = \nu_{(1,1)} = \nu_{(2,2)} = \frac{4}{\pi^2}, \qquad \nu_{(1,2)} = -\frac{4}{\pi^2}, \tag{153}$$

and

$$\mathbb{E}_y[\lambda_1(y)^3] = -\mathbb{E}_y[\lambda_2(\mathrm{y})^3] = \frac{8}{\pi^3}\sum_{y=\pm 1} y^3 = 0. \tag{154}$$

Leveraging eq. (142), the overlap matrices $\boldsymbol{M}$ and $\boldsymbol{Q}$ at convergence satisfy

$$\frac{\|\boldsymbol{M}\|_{\mathrm{F}}^2}{\operatorname{Tr}(\boldsymbol{Q})} = \begin{cases} 1 - \frac{\pi^2}{4}\alpha^{-1}, & \alpha \geq \pi^2/4 \\ 0, & \alpha < \pi^2/4 \end{cases} \tag{155}$$

**E.2.3** $g(z_1,\ldots,z_p) = \prod_{k=1}^p z_k$

For $p = 2$

$$\mathsf{Z}(y) = \frac{K_0(|y|)}{\pi}, \qquad \mathbb{E}[\mathbf{zz}^T|\mathrm{y}] = \begin{pmatrix} |y|\frac{K_1(|y|)}{K_0(|y|)} & \mathrm{y} \\ \mathrm{y} & |y|\frac{K_1(|y|)}{K_0(|y|)} \end{pmatrix}, \tag{156}$$

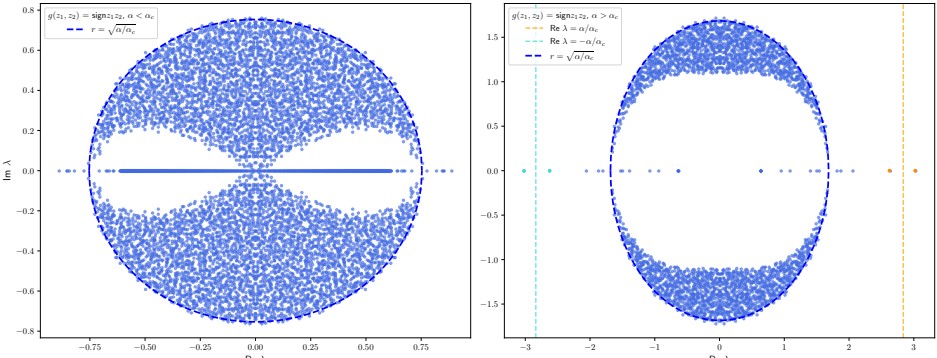

**Figure 5:** Distribution of the eigenvalues (dots) $\lambda \in \mathbb{C}$ of $\boldsymbol{L}$ at finite $n = 5 \cdot 10^3$, for $g(z_1, z_2) = \mathrm{sign}(z_1 z_2)$, $\alpha_c = \pi^2/4$. (**Left**) $\alpha = 1.4 < \alpha_c$. (**Right**) $\alpha = 7 > \alpha_c$. The dashed blue circle has radius equal to $\sqrt{\alpha/\alpha_c}$, *i.e.* the value $\gamma_b$ predicted in Theorem 2.4. The dashed orange vertical line corresponds to $\mathrm{Re}\,\lambda = \alpha/\alpha_c$, the eigenvalue $\gamma_s$ defined in Theorem 2.3. As predicted by the state evolution equations for this problem, two significant eigenvalues (highlighted in orange) are observed near this vertical line. Additionally, one can observe that our framework predicts other two degenerate eigenvalues at $-\gamma_s$, here highlighted in cyan.

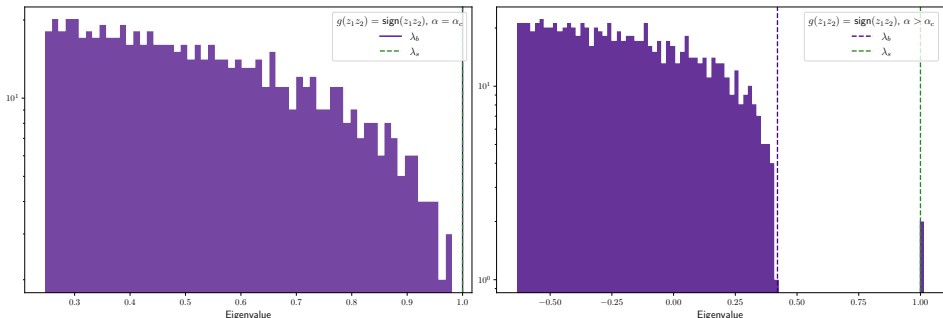

**Figure 6:** Distribution of the eigenvalues of $\boldsymbol{T}$, $d = 10^4$, for the link function $g(z_1, z_2) = \mathrm{sign}(z_1 z_2)$. The critical threshold in $\alpha_c = \pi^2/4$. The distribution is truncated on the left. (**Left**) $\alpha = \alpha_c$. (**Right**) $\alpha = 7 > \alpha_c$. As predicted by the state evolution framework, in this regime we observe two eigenvalues separated from the main bulk, centered around $\lambda_s$ (green vertical line) obtained in Theorem 2.8. The vertical purple line correspond to the value $\lambda_b$ provided in Theorem 2.9 as a bound for the bulk.

where $K_n(\mathrm{y})$ is the modified Bessel function of the second kind. The matrix $\boldsymbol{G}(y)$ is jointly diagonalizable for all $\mathrm{y}$, with eigenvalues $\lambda_1(\mathrm{y}) = |y| \frac{K_1(|y|)}{K_0(|y|)} + \mathrm{y} - 1$ and $\lambda_2(\mathrm{y}) = |y| \frac{K_1(|y|)}{K_0(|y|)} - \mathrm{y} - 1$. Therefore, the eigenvalues of $\mathcal{F}$ are given by[5]

$$\alpha_c^{-1} = \nu_{(1,1)} = \nu_{(2,2)} = \mathbb{E}_\mathrm{y}[\lambda_1(y)^2], \qquad \nu_{(1,2)} = \mathbb{E}_\mathrm{y}[\lambda_1(y)\lambda_1(-\mathrm{y})] < \nu_{(1,1)}, \tag{157}$$

and

$$\mathbb{E}_y[\lambda_1(\mathrm{y})^3] = \mathbb{E}_y[\lambda_2(\mathrm{y})^3]. \tag{158}$$

Leveraging eq. (142), the overlap matrices $\boldsymbol{M}$ and $\boldsymbol{Q}$ at convergence satisfy

$$\frac{\|\boldsymbol{M}\|_\mathrm{F}^2}{\mathrm{Tr}(\boldsymbol{Q})} = \begin{cases} (\alpha - \alpha_c)\left(\alpha + \alpha_c^2 \mathbb{E}_{\mathrm{y}\sim\mathrm{Z}}\left[\lambda_1(y)^3\right]\right)^{-1}, & \alpha \geq \alpha_c \\ 0, & \alpha < \alpha_c \end{cases} \tag{159}$$

If instead $p \geq 3$,

$$\mathrm{Z}(y) = \frac{1}{(2\pi)^{p/2}} G_{0,p}^{p,0}\left(\frac{y^2}{2^p} \,\middle|\, \begin{matrix} 0 \\ 0, 0, \ldots, 0 \end{matrix}\right), \qquad \mathbb{E}[\mathbf{z}\mathbf{z}^T|\mathrm{y}] = (\lambda(\mathrm{y}) + 1)\boldsymbol{I}, \tag{160}$$

where

$$\lambda(y) = 2 G_{0,p}^{p,0}\left(\frac{y^2}{2^p} \,\middle|\, \begin{matrix} 0 \\ 0, 0, \ldots, 0, 1 \end{matrix}\right) \Big/ G_{0,p}^{p,0}\left(\frac{y^2}{2^p} \,\middle|\, \begin{matrix} 0 \\ 0, 0, \ldots, 0 \end{matrix}\right) - 1, \tag{161}$$

---

[5]In what follows, we use $\lambda_1(\mathrm{y}) = \lambda_2(-\mathrm{y})$, the parity of $\mathrm{Z}(y)$ and the symmetry of the integration domain.

and the previous expression are written in terms of the Meijer $G$-function. Therefore $\alpha_c^{-1} = \mathbb{E}_y[\lambda(y)^2]$ and, leveraging eq. (143), we obtain

$$\frac{\|\boldsymbol{M}\|_F^2}{\text{Tr}(\boldsymbol{Q})} = \begin{cases} (\alpha - \alpha_c)\left(\alpha + \alpha_c^2 \mathbb{E}_{y \sim Z}\left[\lambda(y)^3\right]\right)^{-1}, & \alpha \geq \alpha_c \\ 0, & \alpha < \alpha_c \end{cases} \tag{162}$$

### E.3 A non-jointly diagonalizable case: $g(z_1, z_2) = z_1/z_2$

If the matrix $\boldsymbol{G}(y)$ is not jointly diagonalizable $\forall y$, there is not a general simplification for equations (61,62), and each example needs to be treated separately.
In this section we consider the Gaussian multi-index model with link function $g(z_1, z_2) = {}^{z_1}\!/_{z_2}$.

$$Z(y) = \frac{1}{2\pi} \int_{\mathbb{R}^2} e^{-1/2(z_1^2 + z_2^2)} \delta\left(y - \frac{z_1}{z_2}\right) \mathrm{d}z_1 \mathrm{d}z_2 \tag{163}$$

$$= \frac{1}{2\pi} \int_{\mathbb{R}^2} |z_2| e^{-1/2(s^2 z_2^2 + z_2^2)} \delta\left(y - s\right) \mathrm{d}s \mathrm{d}z_2 \tag{164}$$

$$= \frac{1}{2\pi} \int_{\mathbb{R}} |z| e^{-1/2(y^2 + 1)z^2} \mathrm{d}z = \frac{1}{\pi(y^2 + 1)} \tag{165}$$

In order to verify that both directions are not *trivial*, we need to compute $\mathbb{E}[\mathbf{z}|y]$ and verify that is zero almost surely over $y \sim Z$:

$$\mathbb{E}_{\mathbf{z}}[\mathbf{z}|y] \propto \int_{\mathbb{R}^2} \boldsymbol{z} e^{-1/2(z_1^2 + z_2^2)} \delta\left(y - \frac{z_1}{z_2}\right) \mathrm{d}z_1 \mathrm{d}z_2 \tag{166}$$

$$= \int_{\mathbb{R}} \begin{pmatrix} yz \\ z \end{pmatrix} |z| e^{-1/2(y^2 + 1)z^2} \mathrm{d}z = \boldsymbol{0}, \tag{167}$$

where the last equality is the result of the integral of an odd function over a symmetric domain. In order to study the perfomance of the spectral method, we compute

$$\mathbb{E}[\mathbf{z}\mathbf{z}^T | y] = \frac{1}{2\pi Z(y)} \int_{\mathbb{R}^2} \boldsymbol{z}\boldsymbol{z}^T e^{-1/2(z_1^2 + z_2^2)} \delta\left(y - \frac{z_1}{z_2}\right) \mathrm{d}z_1 \mathrm{d}z_2 \tag{168}$$

$$= \frac{1}{2\pi Z(y)} \begin{pmatrix} y^2 & y \\ y & 1 \end{pmatrix} \int_{\mathbb{R}} |z|^3 e^{-1/2(y^2 + 1)z^2} \mathrm{d}z \tag{169}$$

$$= \frac{1}{1 + y^2} \begin{pmatrix} y^2 - 1 & 2y \\ 2y & 1 - y^2 \end{pmatrix} + \boldsymbol{I}. \tag{170}$$

The eigenpairs of $\boldsymbol{G}(y)$ are $\lambda_1(y) = 1$, with eigenvector $(y, 1)^T$, and $\lambda_2(y) = -1$ with eigenvector $(-1, y)^T$, which depends on $y$. Considering a generic $\boldsymbol{M} = \begin{pmatrix} m_1 & m_2 \\ m_3 & m_4 \end{pmatrix}$, we have that

$$\mathcal{F}(\boldsymbol{M}) = \frac{\text{Tr}(\boldsymbol{M})}{2} \boldsymbol{I} + \frac{m_2 - m_3}{2} \begin{pmatrix} 0 & -1 \\ 1 & 0 \end{pmatrix}, \tag{171}$$

therefore, the eigenpairs of $\mathcal{F}$ are

$$\nu_1 = 1, \ \boldsymbol{M}_1 = \boldsymbol{I}; \qquad\qquad \nu_2 = 0, \ \boldsymbol{M}_2 = \begin{pmatrix} 1 & 0 \\ 0 & -1 \end{pmatrix}; \tag{172}$$

$$\nu_3 = 0, \ \boldsymbol{M}_3 = \begin{pmatrix} 0 & 1 \\ 1 & 0 \end{pmatrix}; \qquad \nu_4 = -1, \ \boldsymbol{M}_4 = \begin{pmatrix} 0 & -1 \\ 1 & 0 \end{pmatrix}, \tag{173}$$

and $\alpha_c = 1$. Moreover, one could easily verify that $\mathcal{G}(\boldsymbol{M}_1) = \boldsymbol{0}$. The overlap of the spectral estimator with the signal is therefore $\boldsymbol{M} \propto \boldsymbol{I}$, and, from the state evolution eq. (62) at convergence, we have that

$$\frac{\|\boldsymbol{M}\|_F^2}{\text{Tr}(\boldsymbol{Q})} = \begin{cases} 1 - \alpha^{-1}, & \alpha \geq 1 \\ 0, & \alpha < 1 \end{cases} \tag{174}$$

where we leverage the symmetry of $\boldsymbol{Q}$ in eq. (171) to write $\mathcal{F}(\boldsymbol{Q}) = 2^{-1}\text{Tr}(\boldsymbol{Q})\boldsymbol{I} \implies \text{Tr}(\mathcal{F}(\boldsymbol{Q})) = \text{Tr}(\boldsymbol{Q})$.

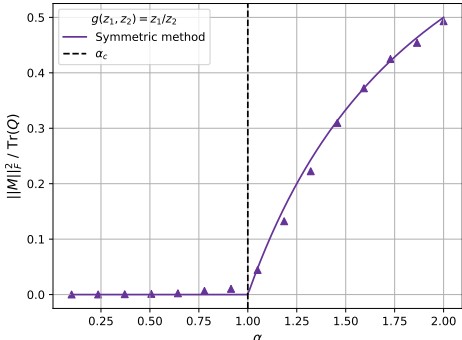

**Figure 7:** Overlap $\|M\|_F^2/\mathrm{Tr}(Q)$ as a function of the sample complexity $\alpha$. The dots represent numerical simulation results, computed for $n = 5000$ (for the asymmetric method) or $d = 5000$ (for the symmetric method) and averaging over 10 instances. The link function is $g(z_1, z_2) = z_1/z_2$. Solid lines are obtained from state evolution predictions. Dashed vertical line at $\alpha_c = 1$.

## F  Details on examples - Symmetric spectral method

In all the considered examples with $p \geq 2$, the matrix $\mathbb{E}[\mathbf{z}\mathbf{z}^T|\mathsf{y} = y]$ admits a unique orthonormal basis of eigenvectors independent of $y$. Therefore, the state evolution equations can be significantly simplified following the same considerations applied in Appendices C.3 and E.2, to which we refer for the notation adopted in this appendix. Additionally, for all these examples, the eigenvalues $\lambda_k(y)$ of $G(y)$ satisfy the additional conditions

$$\forall k, h \in [\![p]\!], \quad \lambda_k(y) = \lambda_h(y) \text{ or } \begin{cases} \lambda_k(y) = \lambda_h(-y), \\ \mathsf{Z}(y) \text{ even and defined on a symmetric domain} \end{cases} \tag{175}$$

so that

$$\forall k, h \in [\![p]\!], \quad \mathbb{E}_\mathsf{y}[\lambda_k(\mathsf{y})^2] = \mathbb{E}_\mathsf{y}[\lambda_h(\mathsf{y})^2] \geq \mathbb{E}_\mathsf{y}[\lambda_k(\mathsf{y})\lambda_h(\mathsf{y})]. \tag{176}$$

It is easy to verify that these conditions implies that $V = I$, and the state evolution admits stable fixed point $M, Q \propto I$, where the proportionality constants can be numerically computed through one-dimensional integrals, expressed in terms of $\lambda_1(y)$ (the choice of the eigenvalue is arbitrary in this setting) and given in eq. (39). Additionally, the largest eigenvalue of $T$ is always equal to one and degenerate (see App. C.3).

Proposition 2.12 readily implies that, for all these examples, the matrix $L$ has a correspondent informative subspace of eigenvectors that can be computed from the subspace of leading eigenvectors of $T$ with eigenvalue equal to 1 and hidden in the bulk.

## G  Proof of Proposition 2.12

1. By definition $L\boldsymbol{\omega} = \gamma_L \boldsymbol{\omega}$. Applying on the right of both sides $\hat{G}(\gamma_L I_{np} + \hat{G})^{-1}$

$$\hat{G}(\gamma_L I_{np} + \hat{G})^{-1} L\boldsymbol{\omega} = \gamma_L \hat{G}(\gamma_L I_{np} + \hat{G})^{-1}\boldsymbol{\omega}. \tag{177}$$

Recalling the definition of $L$ (11), for $i \in [\![n]\!]$, $\mu \in [\![p]\!]$, $k \in [\![d]\!]$

$$\sum_{j,\ell \in [\![n]\!]} \sum_{\nu \in [\![p]\!]} \left[\hat{G}\left(\gamma_L I_{np} + \hat{G}\right)^{-1}\right]_{(i\mu),(j\nu)} \left[\mathbf{X}\mathbf{X}^T\right]_{j\ell} \left[\hat{G}\boldsymbol{\omega}\right]_{(\ell\nu)} = \left[\hat{G}\boldsymbol{\omega}\right]_{(i\mu)}$$

$$\tag{178}$$

$$\implies \sum_{\nu \in [\![p]\!]} \left[G(\mathsf{y}_i)\left(\gamma_L I_p + G(\mathsf{y}_i)\right)^{-1}\right] \sum_{h \in [\![d]\!]} \mathrm{X}_{ih} \underbrace{\sum_{j \in [\![n]\!]} \mathrm{X}_{jh}\left[\hat{G}\boldsymbol{\omega}\right]_{(j\nu)}}_{=w_{(h\nu)}} = \left[\hat{G}\boldsymbol{\omega}\right]_{(i\mu)}$$

$$\tag{179}$$

$$\stackrel{\mathbf{X}^T}{\Longrightarrow} \sum_{i \in [\![n]\!]} X_{ik} \sum_{\nu \in [\![p]\!]} \left[ \boldsymbol{G}(\mathrm{y}_i) \left( \gamma_{\boldsymbol{L}} \boldsymbol{I}_p + \boldsymbol{G}(\mathrm{y}_i) \right)^{-1} \right] \sum_{h \in [\![d]\!]} X_{ih} w_{(h\nu)} = w_{(i\mu)} \tag{180}$$

$$\Longrightarrow \boldsymbol{T}_{\gamma_{\boldsymbol{L}}} \boldsymbol{w} = \boldsymbol{w} \tag{181}$$

2. Defining $\boldsymbol{\omega} := (\boldsymbol{I}_{np} + \hat{\boldsymbol{G}})^{-1} \mathrm{vec}\left( \boldsymbol{X} \mathrm{mat}(\boldsymbol{w}) \right)$, we have that, for $i \in [\![n]\!], \mu \in [\![n]\!]$

$$[\boldsymbol{L}\boldsymbol{\omega}]_{(i\mu)} = \sum_{j \in [\![n]\!]} \sum_{\nu \in [\![p]\!]} \left( [\mathbf{X}\mathbf{X}^T]_{ij} - \delta_{ij} \right) \left[ \boldsymbol{G}(\mathrm{y}_j) \left( \boldsymbol{I} + \boldsymbol{G}(\mathrm{y}_j) \right)^{-1} \right]_{\mu\nu} \sum_{h \in [\![d]\!]} X_{jh} w_{(h\nu)} \tag{182}$$

$$= \sum_{h \in [\![d]\!]} X_{ih} [\boldsymbol{T}\boldsymbol{w}]_{(h\mu)} - [\hat{\boldsymbol{G}}\boldsymbol{\omega}]_{i\mu} \tag{183}$$

$$= \left[ \left( \gamma_{\boldsymbol{T}} (\boldsymbol{I}_{np} + \hat{\boldsymbol{G}}) - \hat{\boldsymbol{G}} \right) \boldsymbol{\omega} \right]_{(i\mu)} \tag{184}$$