# OpenReview forum: "Optimal Spectral Transitions in High-Dimensional Multi-Index Models"
_NeurIPS.cc/2025/Conference — NeurIPS 2025 poster_

### Official Review · Reviewer_suyS · 2025-06-27

**Clarity:** 2
**Significance:** 3
**Originality:** 2
**Rating:** 4
**Confidence:** 3

**Summary:**

The papers studies spectral estimators for high-dimensional multi-index models. Two different spectral estimators are derived, based on linearization of an Approximate Message Passing (AMP) algorithm. These are computed via the principal eigenvector of an asymmetric and a symmetric data matrix, respectively. The weak recovery threshold for the estimators is characterized and shown to coincide with the  “optimal” threshold derived from a fixed point stability condition. Numerical results illustrate the performance of the spectral estimators.

**Questions:**

Questions for the authors are listed in the comments above.

**Ethical Concerns:**

["NO or VERY MINOR ethics concerns only"]

**Final Justification:**

The problem studied is significant and the results are interesting, but the paper is dense and hard to read. An important, related paper also needs to be cited. I have updated my score based on the authors' rebuttal, based on the understanding that these points will be addressed in the revised version.

**Limitations:**

Yes

**Quality:**

2

**Strengths And Weaknesses:**

Strengths:

- The problem studied is interesting and relevant.
- The derived spectral estimators are easily implementable, and the precise characterization of the phase transition thresholds is nice.

Weaknesses:

- The paper is heavy on notation and tersely written. I had to read through multiple times to interpret the main results. Some of this is unavoidable due to the nature of the problem and the result, but the authors could give more high-level explanations and also be more precise with their claims. For instance, it was not clear why the reconstruction threshold was claimed to be optimal? Presumably it because it is also the value under which the trivial fixed point of state evolution is a stable one. Is this a standard definition of optimal threshold?

- Related to the above, I found it hard to interpret Proposition 2.11 Please add a few sentences explaining what it means.

- What is the behavior of the spectral estimators for models that do not satisfy Eq. (6). I suppose when (6) is not satisfied, the problem is easier, but it would be good to know how the estimators perform.

- A recent work, presumably independent and concurrent, studies the high-dimensional asymptotics of spectral estimators (https://arxiv.org/abs/2502.01583). Please comment on how the results and the techniques compare with the ones in that paper.

Typos and minor comments:

- The dimensions don’t seem to match in Eq. (5).
-  Typos: line 115 (throughout), line 118 (matrix)
- Please explain the notation “mat” in (10) and (12). I could guess from context but should be made clear
- In addition to [4], there are other papers in which AMP for multi-index models has been previously studied, e.g. "Matrix Inference and Estimation in Multi-Layer Models" and "Mixed Regression via Approximate Message Passing". Although these papers don't use the multi-index terminology, they study generalized linear models with matrix valued signals, which are the same models as those studied this paper.

---

> ### Author Rebuttal · Authors · 2025-07-30
>
> We thank the reviewer for the careful reading of our work and for acknowledging the relevance of the problem and the usefulness of the proposed spectral estimators. We address each point below and will incorporate the suggested clarifications in the revised version.
>
> ### Weaknesses
>
> 1. We appreciate the feedback and agree that the manuscript could benefit from improved high-level explanations. In the revised version, we will revise the exposition at the start of Sections 2 and 3 to improve accessibility. Regarding the notion of ``optimality'': the threshold $\alpha_c$ is rigorously established in~[18] as the *information-computation threshold* for weak recovery using efficient algorithms. Below $\alpha_c$, no polynomial-time algorithm is known to correlate with the signal $W_\star$; above it, Generalized AMP achieves weak recovery. We restate this in Section 1 and use Lemma 1.2 to recall the definition. Our main result shows that the proposed spectral estimators achieve non-trivial correlation with the signal precisely when $\alpha > \alpha_c$, matching this threshold and thereby achieving optimality in the same well-established sense.
>
> 2.  Proposition 2.11 captures an interesting relationship between the spectra of the symmetric and asymmetric matrices $L$ and $T$. In particular, it shows that an eigenvalue of one in one matrix implies an eigenvalue of one in the other, and that the corresponding eigenvectors define the same subspace estimator. As discussed in Appendix C.3, this implies that even when only one informative eigenvalue appears to separate, additional signal may still be hidden within the bulk. This phenomenon was previously observed in the single-index setting [10] and persists in the more general multi-index models.
>
> 3. In principle, it is possible to utilize the same framework from this manuscript to characterize the overlap of the two spectral estimators for problems not satisfying Eq.(6). However, this does not guarantee optimality, which in this case means achieving weak recovery $\forall \alpha > 0$, as the spectral methods are specifically designed under the condition expressed by Eq.(6). In fact, the linearization of the optimal GAMP in Eq.(14) would not correspond to a power iteration of the matrices $L$ and $T$ if Eq.(6) is not satisfied. The most trivial example is the linear model $g(z) = z$, with $p = 1$. In this case, $\mathsf{Z}(y) = e^{-y^2/2}/\sqrt{2\pi}$, $G(y) = y^2 - 1$, and the weak recovery threshold for the spectral method would be
> $$
> \tilde \alpha_c := \left(\mathbb{E}_{y \sim \mathsf{Z}}[(y^2 - 1)^2]\right)^{-1} = \frac{1}{2},
> $$
> while a single step of the optimal GAMP [18] would achieve weak recovery $\forall \alpha > 0$.
>
> 4. We thank the reviewer for pointing out the concurrent and independent work [KZM25]. While both works address the weak recovery threshold for spectral methods, the techniques and scope differ significantly.
>
> The spectral method proposed in [KZM25] consists in the diagonalization of a $d \times d$ matrix $D$, which recover at most a one-dimensional subspace in $\mathbb{R}^p$. Using the notation from our paper, their "optimal" threshold is defined as
>
> $$
> \delta_c^{-1} := \max_{u \in \mathbb{R}^p, \lVert u \rVert = 1} \mathbb{E}_{y \sim \mathsf{Z}} \left[ (u^\top G(y) u)^2 \right]
> $$
>
> and satisfies
>
> $$
> \delta_c^{-1} \leq \max_{u \in \mathbb{R}^p, \lVert u \rVert = 1} \mathbb{E}_{y \sim Z} \lVert G(y) u u^\top G(y) \rVert_F
> $$
>
> $$
> \delta_c^{-1} \leq \max_{M \succeq 0, \lVert M \rVert_F = 1} \mathbb{E}_{y \sim \mathsf{Z}} \lVert G(y) M G(y) \rVert_F = \alpha_c^{-1}
> $$
>
> $$
> \implies \delta_c \geq \alpha_c
> $$
>
>
> A notable example where the two thresholds differ is the model $g(z_1, z_2) = z_1 / z_2$, for which $\alpha_c = 1$ (see Section 3), while one can easily show that $\delta_c = 2$.
>
> The authors note in Remark 4.1 that their threshold coincides with the one in [18], and hence with the one in our paper, if $G(y)$ is simultaneously diagonalizable for all $y$. For this same subclass of models, as discussed below our Conjecture 2.10, the symmetric spectral method can be interpreted as the diagonalization of $p$ matrices that share the same structure as the matrix $D$ in [KZM25]. In this case, we provide a fully rigorous proof of Conjecture 2.10 in Appendix C.3. We will add a remark on this parallel interesting work in the revised version.
>
>
> [KZM25] Kovačević, Zhang, Mondelli. "Spectral Estimators for Multi-Index Models: Precise Asymptotics and Optimal Weak Recovery". 2025
> ### Typos and minor comments
> - We thank the reviewer for raising the typos in eq. (5) and on lines 115, 118. We will fix them in the revised version.
> - The explanation of the notations $\operatorname{vec}$ and $\operatorname{mat}$ can be found on lines 115-117. In particular, we restrict $\operatorname{mat}$ to act on vectors $a\in\mathbb{R}^{m}$, with $m = np$ or $m = dp$ and we define it as $A = \operatorname{mat}(a) \iff a = \operatorname{vec}(A)$.
> - We thank the reviewer for their comment, we are going to include the references in the Introduction of the revised version.

---

> ### Comment · Reviewer_suyS · 2025-08-04
>
> Thank you for the response. Adding more high-level explanation, especially for results like Proposition 2.11, would make the paper much more readable. Some discussion of what happens when Eq. (6) is not satisfied would also be helpful.
>
> On the understanding that these changes will be made in the revised verison, I will update my score accordingly.

---

### Official Review · Reviewer_Dgt3 · 2025-06-30

**Clarity:** 1
**Significance:** 3
**Originality:** 2
**Rating:** 4
**Confidence:** 3

**Summary:**

The paper studies the problem of learning a Gaussian multi-index model from $n$  noisy samples. Here, the unknown $d$ dimensional function intrinsically resides on a p << d dimensional subspace. The goal is to devise algorithms in the high-dimensional regime where $n/d \rightarrow \alpha$ as $d,n \rightarrow \infty$. Based on the linearization of a message passing scheme, two spectral algorithms are proposed for this problem for recovering the unknown subspace. It is shown that both these methods achieve the optimal reconstruction threshold for weak recovery amongst efficient methods.

**Questions:**

1. Is the link function $g$ assumed to be known to the learner?

2. The matrix $W_{\star}$ is assumed to have independent Gaussian columns but can something be said for any arbitrary $W_{*}$ of rank $p$?

3. In the main results, the effect of $p$ is not visible in the bounds. Is it difficult to track this? Also, do the bounds depend polynomially on $p$? Some remarks on this would be helpful.

**Ethical Concerns:**

["NO or VERY MINOR ethics concerns only"]

**Final Justification:**

The authors have clarified the technical questions I had and I am satisfied with the response. However my concern regarding the presentation still remains - the technical details are very hard to parse which makes it difficult to verify their correctness.

**Limitations:**

Yes discussed towards the end

**Quality:**

3

**Strengths And Weaknesses:**

Pros
--------

1. The results in the paper establish weak recovery results for Gaussian multi-index models which generalize existing work for single-index models. In that respect, the results do extend the state-of-art for this problem and would be of interest to researchers in high-dimensional statistics working on similar problems.

Cons
----------
1. Unfortunately, I found the exposition to be a bit hard to follow at some places, despite multiple passes of the text. The issue starts from page 3 onwards where a lot of technical details regarding GAMP is provided, but with the details deferred to the appendix. This makes it difficult to develop good intuition, even for the spectral methods introduced next, which, as the authors note, do appear to be a bit ad hoc.

2.  As noted in the text, the proposed spectral methods are generalizations of existing similar methods for the single-index case. While this is fine in general of course, it does limit the novelty aspect a tad bit. Perhaps the main question is – what are the technical novelties/ difficulties in the analysis compared to the single index case? This doesn’t seem to have been discussed in detail but would make things more transparent for the reader.

---

> ### Author Rebuttal · Authors · 2025-07-30
>
> We thank the reviewer for their careful assessment of our work and their feedback. We address each point below.
>
> ### Weaknesses
> 1. We thank the reviewer for raising this issue. We have chosen not to include the explicit formulation of the optimal GAMP algorithm in the main text, as it plays a role exclusively in the derivation of the spectral methods through its linearization, reported in Eq. (14). Although this derivation brings insight on the method, it could be fully defined without referencing GAMP. We intend to clarify this point in the revised version and use the additional space to provide some intuitive remarks supporting the mathematical expressions.
> 2. We thank the reviewer for this comment and the opportunity to clarify. While the proposed spectral methods reduce to existing ones in the single-index case $(p=1)$, this is by design and reflects a desirable consistency with prior work. We respectfully disagree with the implication that this limits the novelty of our contribution.
>
>    The core contribution of our work lies precisely in going beyond the well-understood single-index case to develop spectral methods that operate—and provably succeed—in the far more intricate multi-index setting $(p>1)$. This generalization is highly non-trivial and introduces a range of new technical challenges that are absent in the scalar case, including matrix-valued state evolution, operator-level analysis, and the spectral behavior of block-structured random matrices. For the asymmetric spectral method in particular, this is the first time such a method has been derived and its asymptotic behavior characterized—even for $p=1$.
>
>     Furthermore, our work provides a first rigorous proof of a spectral phase transition in the multi-index setting (Conjecture 2.10), for a broad class of models. To do so, we had to identify special structures that allow simultaneous diagonalization and then extend the random matrix techniques from [52] to handle the higher-rank regime—a nontrivial technical contribution that required substantial innovation.
>
>     In short, the reduction to known results for $p=1$ is not a limitation, but a necessary and validating boundary case. The bulk of the contribution lies in overcoming the deep theoretical obstacles that arise when moving beyond this case.
>
> ### Questions
> 1. Indeed, our setting assumes the knowledge of $\mathsf P (\cdot | z)$ and consequently of the link function $g(z)$. The reason is that, as emphasized in the title, the focus of the work is on understanding optimal spectral transitions, and as such we restrict ourselves to the information-theoretically optimal setting. While this is a standard assumption in the analysis of spectral methods and AMP, we agree that extending the analysis to account for mismatched or approximate link functions is an interesting direction for future work. This can be readily done within our formalism.
> 2. We thank the reviewer for raising this point. In fact, our results do not depend on $W^\star$
>   being drawn from a Gaussian distribution. The Gaussian prior is used only for deriving the associated GAMP algorithm,  However, once the algorithm is linearized, the behavior of the spectral methods depends only on the subspace spanned by $W^*$, provided it has full rank and normalized Frobenius norm. This is consistent with what is known for $p=1$, where the performance of spectral methods depends only on the norm of the teacher vector and not its detailed distribution.
>
>     That said, it would be very interesting to explore the case where $W^*$  is rank-deficient or structured in other ways. The GAMP framework can be extended beyond separable priors to more general joint distributions. In principle, the same pipeline—AMP derivation, linearization, spectral construction, and state evolution analysis—can be applied in such settings. While technically more involved, this would be a conceptually feasible and valuable extension.
>
> 3. The dependence on $p$ is model-specific and not universal. While our general theorems are stated abstractly, the effect of
> $p$ becomes apparent in specific examples. For instance the critical threshold for the link function $g(z\in\mathbb{R}^p) = ||z||^2/p$ is $\alpha_c = p/2$ and the overlap of the asymmetric estimator is given by $m^2 = \max((\alpha - p/2)/(\alpha+2), 0)$. These examples, included in the manuscript, illustrate how $p$ can directly influence both the sample complexity and the recovery performance.

---

> > ### Comment · Reviewer_Dgt3 · 2025-08-05
> > **Read rebuttal**
> >
> > Thank you for clarifying my questions in detail, I appreciate it. I am fine with the explanations given, especially regarding the novelty aspect. My main concern regarding the presentation however still remains - I found the technical details to be very hard to parse which makes it difficult to verify the correctness of the technical claims. Perhaps this is unavoidable for the GAMP framework, but it does make it difficult for me to raise my score. So I will keep it for now.

---

### Official Review · Reviewer_TdfR · 2025-07-02

**Clarity:** 2
**Significance:** 3
**Originality:** 2
**Rating:** 4
**Confidence:** 3

**Summary:**

This study introduces two new spectral algorithms for weak recovery in Gaussian multi-index models. The authors review related literature, noting that while multi-index models have been thoroughly analyzed in the context of learning with neural networks, they approach the problem from the perspective of random matrix theory, where the sample complexity required to recover the relevant subspace remains an open question. Previous works conjectured a threshold but achieved it only with algorithms that relied on additional side information. The authors claim that the present work closes this gap and conjecture that the proposed spectral algorithms achieve weak recovery at the optimal sample complexity threshold. They support this claim through an analysis of the state evolution of an associated message passing scheme and extensive simulations.

**Questions:**

### Questions
- There appears to be a mistake in Definition 1.1, as $W_* \in \mathbb{R}^{d \times p}$ cannot be multiplied on the right by $v \in \mathbb{R}^d$. It is also unclear how to correct the expression, as there are no accompanying comments on the definition. Should $v \in \mathbb{R}^p$, or is an additional transpose operator missing inside equation (5)? Alternatively, perhaps the intention is to define weak recovery for subspaces of $W_*$, with $V \subset \mathbb{R}^p$?
- In Figure 2, why are the orange, green, and purple lines not present for $\alpha < \alpha_c$? Are they overlapping with the blue line? If so, I would suggest using dashed or dotted lines to make overlapping curves visible.

### Minor Remarks
- For semipositive matrices, there is inconsistency in the notations: the + is sometimes subscripts and sometimes superscript.
- Lines 111–112: repetition of the word "functions."
- Line 118: "matrix," not "matix."
- There is inconsistency in the notation for the identity matrix; sometimes the dimension is included as a subscript and sometimes omitted. I suggest always specifying the dimension for clarity.

**Ethical Concerns:**

["NO or VERY MINOR ethics concerns only"]

**Final Justification:**

Besides my initial positive impression on the contribution this study provides to the theory of multi-index models, the rebuttal from the authors made me appreciate other contributions towards conjecture 2.10 that did not find space in the main text.
The links with reference [18] were explained in more details, clarifying better the difference of the two works.

**Limitations:**

Addressed

**Paper Formatting Concerns:**

No concerns

**Quality:**

3

**Strengths And Weaknesses:**

### Strengths
- Spectral methods are important both in random matrix theory and high-dimensional statistics. The introduction of a well-performing spectral method for a topical model represents a significant contribution.
- Although the paper does not provide a rigorous proof of the optimality of the algorithms, the analysis of the state evolutions of the message-passing schemes, together with the numerical simulations, offers solid evidence for the optimality of the introduced spectral algorithms.

### Weakness
The work relies heavily on reference [18], which also focused on the study of a message-passing algorithm. It is not clear how closely related the algorithm introduced in the present paper is to the one discussed in [18]. This connection should be clarified further

---

> ### Author Rebuttal · Authors · 2025-07-30
>
> We thank the reviewer for their careful assessment and positive evaluation. We would like to clarify that for a broad class of models—including several relevant examples detailed after Conjecture 2.10—we are able to rigorously validate the conjectured spectral phase transition. Specifically, we complement the state evolution analysis with a formal random matrix theory characterization of the spectrum of the matrix
> $T$, thus fully proving Conjecture $2.10$ in these cases (see Appendix $C.3$ and Theorem $C.1$).
> ### Weaknesses
> We appreciate the opportunity to clarify this point. Reference [18] indeed establishes the computational threshold for weak recovery in Gaussian multi-index models via an optimal Generalized Approximate Message Passing (GAMP) algorithm. However, a crucial limitation of that work is that its algorithm requires *an informative initialization*—without which it remains trapped in a non-informative fixed point. Overcoming this limitation is one of the key motivations behind our approach.
>
> Our contribution is fundamentally different. We construct *two spectral methods*, derived from the *linearization of the AMP dynamics in [18] around the uninformative fixed point*, that require *no side information*. In doing so, we replace a complex iterative AMP scheme with simple, off-the-shelf spectral algorithms that are both practical and theoretically insightful.
>
> While inspired by the structure of [18], the proposed methods are *algorithmically distinct*: they stem from new, explicitly defined linearized AMP schemes (Definitions 2.1 and 2.6) and are analyzed via their *own state evolution equations*, which serve not to track an estimator, but to characterize the *asymptotic behavior of the top eigenvectors* of the associated matrices $L$ and $T$.
>
> This perspective allows us to *close the gap left open in [18]*: we achieve weak recovery at the same critical sample complexity $\alpha_c$, but *without any prior knowledge of the signal*—purely via spectral methods.
>
> ### Questions
> 1. Indeed, this is a typo, as $v\in V\subset \mathbb{R}^p$. We are going to fix this in the revised version.
>  2. The lack of theoretical predictions for $\alpha < \alpha_c$ is due to the absence of a solution to $\nu^{\tilde{\mathcal{F}}} (a) = \alpha^{-1}$ (Theorem 2.9) for the considered model. In fact, as mentioned in Section 2.2, a solution is guaranteed to exist only for $\alpha > \alpha_c$. Although this partial analysis limitation, the state evolution for the GAMP in Definition 2.6 allows us to predict that an informative eigenvalue of $T$ separates from a bulk of eigenvalues for $\alpha>\alpha_c$, which is the regime of interest. The absence of informative eigenvalues at smaller sample complexities is guaranteed by the results in [18].
>
> ### Minor Remarks
> - We thank the reviewer for pointing the small inconsistencies and typos. We are going to take care of them in the revised version.

---

> > ### Comment · Reviewer_TdfR · 2025-08-02
> >
> > Thanks for the detailed answer, especially for pointing out the additional section in appendix C in which you consider the special case of joint diagonalization in which conjecture 2.10 that is fully proved. I believe this is an important contribution that should be stressed more in the main paper. I will updated my score accordingly.

---

### Official Review · Reviewer_cfKb · 2025-07-03

**Clarity:** 3
**Significance:** 4
**Originality:** 3
**Rating:** 5
**Confidence:** 3

**Summary:**

The authors present an AMP-based analysis of two spectral algorithms for Gaussian multi-index GLMs. Based on an analysis of the fixed points of GAMP with linear denoisers, two different spectral methods are provided, based on the eigenvalues (or singular values, depending on the approach) of two specially de- signed matrices that depend on observed data and the (known) likelihood for the observations given the covariates.

For both methods, a critical threshold for the sample complexity of the problem below which the spectral methods achieve asymptotically zero overlap with the ground-truth weights. Conversely, it is shown that above thresholds the proposed methods achieve weak recovery. The outputs of the two methods are related to each other in an exhaustive fashion.
For both approaches it is conjectured that the spectral distribution of the matrices used for the two spectral methods converges weakly to some distribution supported on a compact disk, with the exception of some outliers correlated with the weight matrices. These conjectures are supported by convincing numerical experiments.

Overall, this paper is well-written and introduces novel spectral methods for multi-index models, thus generalizing previously known results about GLMs. On the technical side, relatively standard AMP techniques are used and the degree of novelty is somewhat reduced by the lack of proof for the spectral distribution of the operators described in the proposed algorithm. Nonetheless, I think this is a solid paper.


Comments:
- The very last part of the section on numerical experiments is dedicated to providing some closed forms for the limiting overlap and critical sample complexity for specific link functions. This part is very cramped and quite unpleasant to read. I think it’d be better to place all expressions in the same appendix in which they’re derived and just have a pointer in the main text. Maybe use the extra space to expand on the description of the numerics and add the form of the two operators $F$ and $G$, which are defined in appendix but used in main text

- Equation (14) and the lines immediately after use some $\delta$ as a symbol. According to the claimed notational conventions this should be a scalar, but usage suggests otherwise, and its role is unclear. Furthermore, Kronecker deltas are used in other definitions, making it even more confusing.

- Line 79: independant -> independent

**Questions:**

see summary.

**Ethical Concerns:**

["NO or VERY MINOR ethics concerns only"]

**Final Justification:**

I am keeping my score because the paper is solid.

**Limitations:**

yes.

**Paper Formatting Concerns:**

no concerns.

**Quality:**

4

**Strengths And Weaknesses:**

see summary.

---

> ### Author Rebuttal · Authors · 2025-07-30
>
> We thank the reviewer for their feedback, appreciation and suggestions. The typos will be corrected in the revised version.
>
> - We intend to improve the presentation of the results for specific link functions and clarify the form of the mentioned operators leveraging the additional page in the camera-ready version.
>
> - The $\delta$ in $\delta W$ and $\delta \Omega$ is not meant as a multiplication, but to denote a variation, used to distinguish the linearized algorithm rates from the GAMP ones. We will clarify the notation in the revised version.
>
> Furthermore, we would like to stress that a fully rigorous proof for our Conjecture 2.10 is included in Appendix C.3. As mentioned on lines 228-240, this formalisation, based on random matrix theory results, concerns a subset of models including several problems of interest. We will clarify this matter in the revised version.

---

> > ### Comment · Reviewer_cfKb · 2025-08-05
> >
> > I appreciate the authors responses. Overall I feel this is a solid paper. But I am disappointed to learn that there was relevant related work (e.g., the reference [KZM25]) that was not mentioned. If the authors mention this as parallel work in their initial submission then it becomes strength (e.g., it appears there is lots of interest in this topic). If the authors wait for a reviewer to point it out then it reflects poorly.

---

> > > ### Author Response · Authors · 2025-08-05
> > >
> > > We thank for the reviewer feedback. As discussed in the reply to Reviewer *suyS*, [KZM25] is a concurrent work, and due to an ongoing discussion with the authors of this work during the period of the NeurIPS submission, we have postponed adding this acknowledgement to make sure everyone agrees it fairly acknowledges their contribution with respect to ours. We have since agreed on a mutual acknowledgement, and we can reassure the reviewer it will be in the revised version of the submission. We completely agree with the point of view that this strengths the work rather than diminishes it.

---

### Decision · Program_Chairs · 2025-09-17

**Decision:**

Accept (poster)

**Comment:**

The paper introduces novel spectral methods for weak recovery in Gaussian multi-index GLMs, generalizing prior results for GLMs and single-index models. These methods are based on an AMP-based analysis, deriving two spectral algorithms from the eigenvalues or singular values of specially designed matrices. The authors characterize a critical sample complexity threshold below which the methods achieve asymptotically zero overlap with ground-truth weights, and above which they achieve weak recovery. It is conjectured that the spectral distribution of the matrices converges weakly to a distribution supported in a disk, with some informative outliers.

This represents a significant contribution in both random matrix theory and high-dimensional statistics. The precise characterization of phase transition thresholds and the ease of implementation of the derived spectral estimators are also noted. While a rigorous proof of optimality is lacking, the analysis of state evolutions of associated message-passing schemes, coupled with extensive numerical simulations, provides solid evidence for the optimality of the proposed spectral algorithms.

Conversely, a recurring weakness across reviews is the paper's dense and occasionally difficult-to-follow exposition, particularly from page 3 onwards where technical details are deferred to the appendix without sufficient high-level explanation. This makes it challenging to develop intuition for the introduced spectral methods. Reviewers also point out that the reliance on and generalization of existing methods for the single-index case somewhat limits the novelty, and suggest a clearer discussion of the technical novelties and difficulties compared to the single-index case. Furthermore, the paper should clarify the relation to prior work [18] and clarify the connection between the algorithms.

With the understanding that the authors will make a good-faith effort to improve the presentation, which confounded even experts in AMP, and implement the specific suggestions of the reviewers, accept.